# BMP4 drives primed to naïve transition through PGC-like state

Shengyong Yu[1,9], Chunhua Zhou[1,9], Jiangping He [2,9], Zhaokai Yao [3], Xingnan Huang[4], Bowen Rong[5], Hong Zhu[1,6], Shijie Wang[7], Shuyan Chen[1,6], Xialian Wang[7], Baomei Cai[2], Guoqing Zhao[2], Yuhan Chen[3], Lizhan Xiao[2], He Liu[2], Yue Qin[1,6], Jing Guo[1], Haokaifeng Wu[1], Zhen Zhang[1], Man Zhang [2], Xiaoyang Zhao [3], Fei Lan[5], Yixuan Wang[1,8], Jiekai Chen [1,2], Shangtao Cao [2✉], Duanqing Pei [4✉] & Jing Liu [1,2✉]

Multiple pluripotent states have been described in mouse and human stem cells. Here, we apply single-cell RNA-seq to a newly established BMP4 induced mouse primed to naïve transition (BiPNT) system and show that the reset is not a direct reversal of cell fate but goes through a primordial germ cell-like cells (PGCLCs) state. We first show that epiblast stem cells bifurcate into *c-Kit*+ naïve and *c-Kit*− trophoblast-like cells, among which, the naïve branch undergoes further transition through a PGCLCs intermediate capable of spermatogenesis in vivo. Mechanistically, we show that DOT1L inhibition permits the transition from primed pluripotency to PGCLCs in part by facilitating the loss of H3K79me2 from *Gata3/6*. In addition, *Prdm1/Blimp1* is required for PGCLCs and naïve cells, while *Gata2* inhibits PGC-like state by promoting trophoblast-like fate. Our work not only reveals an alternative route for primed to naïve transition, but also gains insight into germ cell development.

[1] Center for Cell Lineage and Development, Guangzhou Institutes of Biomedicine and Health, Chinese Academy of Sciences, Guangzhou 510530, China.
[2] Center for Cell Lineage and Atlas, Bioland Laboratory, 510005 Guangzhou, China. [3] State Key Laboratory of Organ Failure Research, Department of Developmental Biology, School of Basic Medical Sciences, Southern Medical University, Guangzhou 510515, China. [4] Laboratory of Cell Fate Control, School of Life Sciences, Westlake University, Hangzhou 310024, China. [5] Shanghai Key Laboratory of Medical Epigenetics, State International Co-laboratory of Medical Epigenetics and Metabolism, Institutes of Biomedical Sciences, Fudan University, and Key Laboratory of Carcinogenesis and Cancer Invasion, Ministry of Education, Liver Cancer Institute, Zhongshan Hospital, Fudan University, Shanghai 200032, China. [6] University of Chinese Academy of Sciences, Beijing 100049, China. [7] GMU-GIBH Joint School of Life Sciences, Guangzhou Medical University, Guangzhou 511436, China. [8] School of Life Sciences, University of Science and Technology of China, Hefei 230026, China. [9] These authors contributed equally: Shengyong Yu, Chunhua Zhou, Jiangping He.
✉email: cao_shangtao@grmh-gdl.cn; peiduanqing@westlake.edu.cn; liu_jing@gibh.ac.cn

Pluripotency is defined as the ability to generate all cell types in an individual by a single cell. In mice, it has been well established that two distinct, yet interconvertible, states exist, namely naïve and primed pluripotency represented by ESCs (Embryonic Stem Cells) and EpiSCs (Epiblast Stem Cells)[1–3]. Despite both referred to pluripotency, they differ markedly in morphology, gene expression profile, conditions for maintenance, epigenetic state, chimeric competence, and germline contribution[1]. From an embryological perspective, naïve and primed pluripotency represent the developmental potentials of pre- and post- implantation embryos, respectively[1]. Therefore, a detailed investigation of both states may deepen our understanding of early embryo development.

The interconvertibility between naïve and primed states has been well documented[4–8]. For example, naïve ESCs can be differentiated into primed state closely resembling EpiSCs[9,10]. Likewise, EpiSCs have been converted to the naïve state similar to ESCs[4,5]. Unlike the differentiation of ESCs to the primed state, the reprogramming of EpiSCs towards naïve state has been largely accomplished by over-expressing transcription factors (TFs)[5,11–16], similar to the iPSCs (Induced Pluripotent Stem Cells) pioneered by Yamanaka and colleagues[17]. Recently, we have established a BMP4 induced primed to naïve transition (BiPNT) system without any exogenous TF expression, and show that a 3-day exposure of EpiSCs in BMP4, DOT1L inhibitor and EZH2 inhibitor followed by a 5-day culture in 2iL can convert the majority (~80%) of the primed cells into chimera-competent naïve colonies[18]. Through RNA-seq and ATAC-seq bulk analyses, we demonstrated that many downstream targets of BMP4, including the previously unrecognized Zbtb7 family members, are activated through chromatin remodeling events for the resetting of primed to naïve states[18]. Even though it is still unclear if there are intermediates governing the resetting of primed into naïve state by BMP4 or the precise cell dynamics during this resetting process, as distinct molecular routes have been shown to mediate the acquisition of naïve pluripotency with different TFs and signals[19].

Here we report the cell fate continuum between primed and naïve states generated by BMP4 with intermediate states previously masked in bulk analysis. Specifically, our data reveal a Prdm1-dependent PGCLCs intermediate governing BiPNT.

## Results

**Single-cell atlas of BMP4 induced primed to naïve transition.** We have established a robust BMP4 induced primed to naïve transition (BiPNT) system[18], that can convert ~80% EpiSCs to chimera-competent naïve colonies. To map the cell fate continuum, we performed scRNA-seq from cells that are at D0, 1, 2, 3, 5, and 8 (Fig. 1a). Analysis with t-SNE methods yielded well segregated clusters for cells at each day (Fig. 1b), indicating distinct stages of cell fate transition during BiPNT. Indeed, distinct gene expression profiles can be observed from D0 to D8 using representative genes from primed to naïve state (Fig. 1c).

To further identify the cell populations at two stages, we analyzed single cells from D0–3 at stage 1 in detail and identified that many cells emerged at D3 have naïve pluripotency program genes (Fig. 1d). Among those genes, Klf2, Dppa5a, and Dppa3 are almost exclusively expressed in D3 cluster (Fig. 1d, lower panels). Interestingly, we also found cells in D3 expressing some trophoblast marker genes, such as Plac1, Gata2, Krt8, and Peg10 (Fig. 1d, middle panels). These results suggest that EpiSCs undergo fate transitions towards trophoblast-like and naïve-like pluripotency cells at stage 1.

2iL is a well-known condition for maintaining naïve pluripotent state[20]. When the BMP4-treated cells were switched to 2iL

condition, it is expected that they can be further matured into naïve pluripotency. Indeed, scRNA-seq confirms naïve pluripotent cells between D5 and D8 (Fig. 1e), suggesting that 2iL has further reprogrammed the cells in D3 into naïve state. We can also observe minor populations in both D5 and D8 cultures as trophoblast-like (Fig. 1e).

We then plotted the cells with naïve- and trophoblast-like programs to show that BMP4 induces almost equal numbers for each fate up to D2, but the naïve fate continues to increase while the trophoblast-like fate levels off (Fig. 1f). Apparently that 2iL continues to support the increase of the naïve fate as expected while the trophoblast-like ones begin to diminish at D5 (Fig. 1f). Thus, we can conclude that two primary fates emerge in BMP4-treated EpiSCs and 2iL favors the naïve fate while partially suppressing the trophoblast-like program (Fig. 1f). Gata2, which is highly expressed in trophoblast but not in naïve pluripotent stem cells, plays an important role in trophoblast development[21,22]. To further trace the generation of these two cell fates in vitro, EpiSC cell line carrying both ΔPE-Pou5f1 (also known as Oct4) -GFP[23]/ Gata2-tdTomato reporters was constructed and then undergoes BiPNT (Supplementary Fig. 1a). Results from both fluorescence microscope in situ (Fig. 1g) and FACS (Supplementary Fig. 1b) show that, the naïve signal (ΔPE-Oct4-GFP) and trophoblast signal (Gata2-tdTomato) are incompatible and mutually exclusive to each other, confirming two discrete cell types at the end of BiPNT.

We further analyzed the identity of those trophoblast-like cells by comparing the gene expression profiles of D5/D8 trophoblast-like cells to the published reference dataset[24], and show by Sankey plot that the majority (87.53%) of D5/D8 trophoblast-like cells can be classified into ExE (extra-embryonic) ectoderm, while 7.97% cells are difficult to be classified (5.14%, Rejected; 2.83%, Ambiguous), with very few cells can be classified as ExE endoderm, ExE mesoderm, or other cell types (Supplementary Fig. 1c). Noteworthily, the amnion markers such as Isl1 and Igfbp3 are exclusively expressed in those D5/D8 Elf5 positive trophoblast-like cells (Supplementary Fig. 1d), which can distinguish them from the amnion cells.

**c-Kit marks the naïve branch.** To further resolve the apparent naïve versus trophoblast-like fate choices in D2 and D3, we re-analyzed the scRNA-seq data with Harmony[25], a program designed to resolve branches or trajectories along cell fate continuum. Harmony resolves the cells into two main branches, the trophoblast branch (TB) and naïve branch (NB) as expected (Fig. 2a, b). While the TB is enriched with imprinted genes such as Igf2, Peg10, H19 and trophoblast genes Gata2, Tead3, the NB expresses Dppa5a, Klf2, and Nanog, mostly naïve-related genes (Fig. 2b). Consistent with the naïve- and trophoblast-dichotomy, the TB is enriched with genes from trophoblast-like fate while the NB with those from the naïve-like fate (Fig. 2c).

We then searched for cell surface markers that may distinguish the NB from the TB. Among a group of cell surface markers (Supplementary Fig. 2a), we found that Kit (also known as c-Kit) is a good candidate (Fig. 2d). c-Kit begins to be expressed at D1 and persists in the NB significantly (Fig. 2d). When plotted along the pseudotime, it becomes very clear that c-Kit begins to diverge at early phase of PNT that the NB maintains its expression while the TB diminishes (Fig. 2e), suggesting that c-Kit may help mark the NB. To test this idea, we sorted D3 cells with c-Kit-APC and show that ~50% of D3 cells are c-Kit positive (c-Kit⁺) (Fig. 2f). By replating the sorted c-Kit⁺ and c-Kit⁻ cells separately and culturing them in 2iL, we show that almost all (94.2%) of c-Kit⁺ cells become GFP positive compared to 0.7% for c-Kit⁻ cells (Fig. 2g).

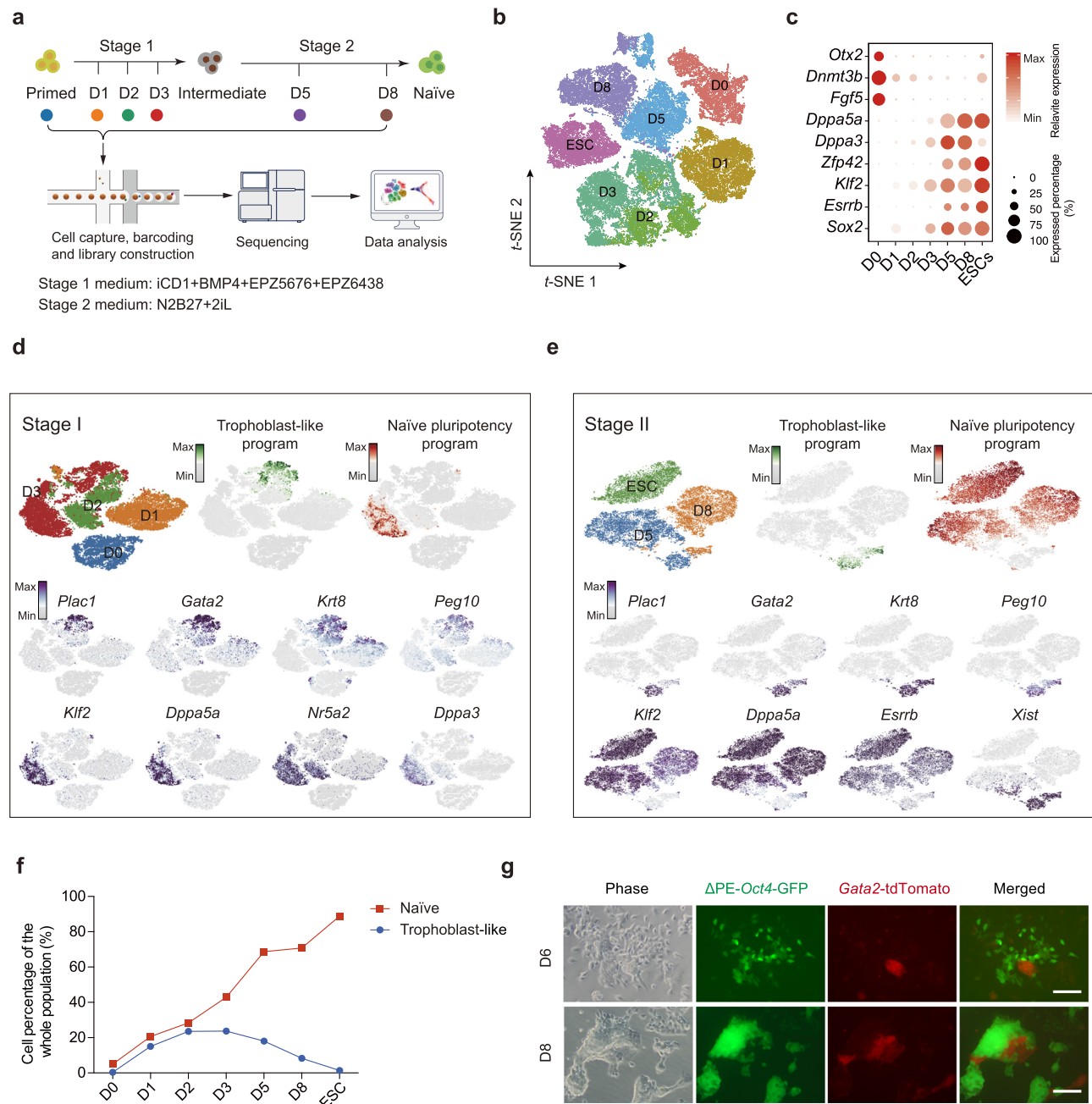

**Fig. 1 Single-cell analysis for BiPNT. a** Flow chart for the scRNA-seq analysis of BMP4 induced Primed-Naïve Transition (BiPNT). scRNA-seq experiment was performed with 10x Genomics. **b** t-SNE projection of all 50406 individual cells during the whole BiPNT process. **c** Expression of genes in 2 different categories at the indicated time points. **d** t-SNE projection of cells and typical gene expressions at stage 1. **e** t-SNE projection of cells and typical gene expressions at stage 2. **f** Percentages of naïve cells and trophoblast-like cells at different time points. **g** Representative images of ΔPE-Oct4-GFP/Gata2-tdTomato EpiSC induced BiPNT at D6 and D8. Scale bars, 100 μm. The experiments were repeated independently three times with similar results.

We further performed RNA-seq to investigate the difference between Day 3 *c-Kit*⁺ and *c-Kit*⁻ population (Fig. 2h), and show that 1102 genes are highly expressed in *c-Kit*⁻ cells, including the trophoblast markers *Plac1*, *Gata2*, and *Krt8*, while 1065 genes are specifically expressed in *c-Kit*⁺ cells including pluripotency genes *Pou5f1*, *Sox2*, *Nanog*, *Esrrb*, *Klf2*, and more interestingly, *T* and *Prdm1* (also known as *Blimp1*) (Fig. 2h). GO (gene ontology) analysis reveal that programs such as embryonic placenta development, endothelial cell migration, and placenta development are highly enriched in *c-Kit*⁻ cells, while those involved in gastrulation, BMP signaling pathway, non-canonical Wnt signaling pathway are highly enriched in *c-Kit*⁺ cells (Fig. 2i).

Given *c-Kit* and *Prdm1* are well-known primordial germ cell (PGC) markers[26], we then detected the expression of other PGC markers in Day3 sorted *c-Kit*⁺ cells by RT-qPCR. PGC genes such as *Dppa3* (also known as *Stella*), *Nanos3* and *Ifitm1/3* are also highly expressed in *c-Kit*⁺ cells, while trophoblast genes such as *Plac1* and *Gata2* are highly expressed in *c-Kit*⁻ cells (Fig. 2j). Consistently, we also detected stronger ATAC-seq signaling in loci for PGC genes such as *Nanog*, *Prdm1*, and *Prdm14* in *c-Kit*⁺ cells, as well as for trophoblast genes such as *Plac1*, *Gata2*, and *Elf5* in *c-Kit*⁻ cells (Supplementary Fig. 2b). Together, these results suggest that *c-Kit* marks the naïve branch quite early among BMP4-treated cells.

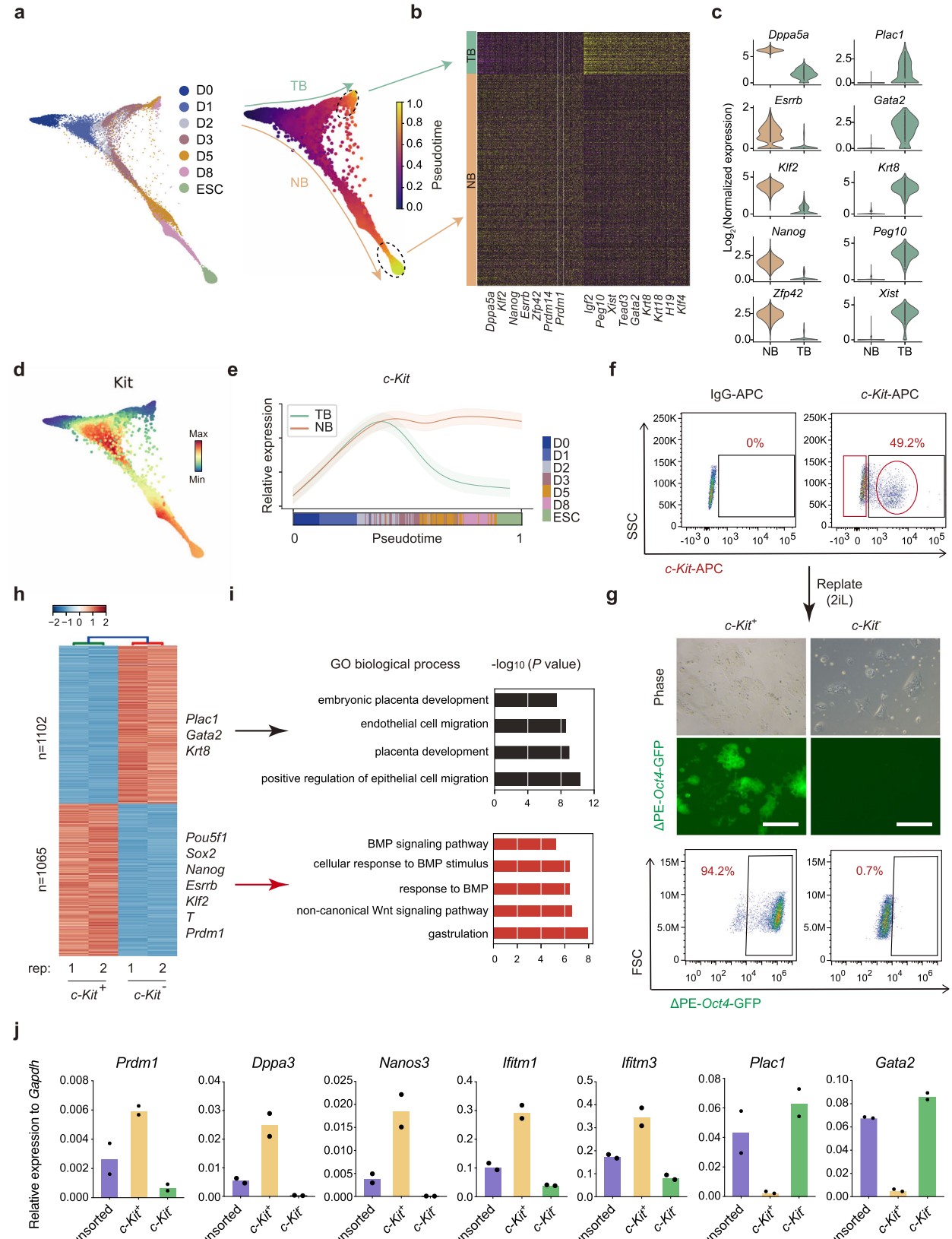

**Activation of a PGC-like program during BiPNT**. The enrichment of PGC markers in Day 3 *c-Kit*⁺ cells (Fig. 2h, j) warrants further analysis. We then generated a pseudotime plot of NB during PNT and found it can be generally divided into four distinct phases (I–IV) (Fig. 3a). The first phase (I) is mainly cells expressing genes enriched in the primed fate, such as *Otx2*,

*Dnmt3b*, and *Fgf5*. The second phase (II) includes cells that appear to express genes enriched for mesoderm like PGC precursor[27], such as *Msx1/2*, *Cdx2*, *T*, and *Prdm1*. The third phase (III) are cells expressing genes enriched in early PGC, such as *Prdm14*, *Tfap2c*, *Dppa3*, *Nanos3*, and *Dnd1*. And the last phase (IV) contains cells with naïve pluripotency features, such as *Klf4*

**Fig. 2 c-Kit is a key marker for successful BiPNT. a** Trajectory reconstruction of all single cells throughout BiPNT, colored by indicated time points. **b** The bifurcating branches were defined as NB (naïve branch) and TB (trophoblast branch) separately. Circles indicate cells at the terminus of the NB and TB, respectively (left). Identification of the differentially expressed genes between NB and TB (right). **c** Violin plot displaying the expression of representative naïve pluripotency genes and trophoblast genes in indicated cell types. **d** Plots showing the expression patterns of cell surface marker *c-Kit* in NB and TB. **e** Expression trends of *c-Kit* along pseudotime during NB and TB specification. **f** *c-Kit* positive and negative cells were separated at Day 3 of BiPNT. **g** Images of ΔPE-*Oct4*-GFP colonies (upper panel) and FCAS result (lower panel) for *c-Kit* positive and negative cells sorted in **f** and replated in stage 2 medium (2iL) for another 5 days. Scale bars, 250 μm. The experiments were performed twice with similar results. **h** Heatmap showing gene expression differences between *c-Kit* positive and negative cell samples by RNA-seq. Representative genes are shown on the right. **i** Gene ontology (GO) analysis for differentially expressed genes in **h**. **j** Expression of PGC and trophoblast markers measured by RT-qPCR in Day 3 cells of BiPNT (unsorted, sorted *c-Kit* positive and negative cells). Data are mean ± s.d., $n = 2$ independent experiments. Source data are provided as a Source Data file.

and *Tbx3* (Fig. 3a). Consistently, heatmap generated with Harmony also indicates a continuous cell fate transition from primed pluripotency, PGC precursor, early PGC, then end with naïve pluripotency along the NB (Fig. 3b).

To distinguish the putative early PGC-like cells (PGCLCs) from naïve pluripotent cells, we detected the expression of KLF4 in ΔPE-*Oct4*-GFP positive cells by immunofluorescence (IF) at Day 6 and Day 8 during BiPNT[28]. Since PGC and naïve pluripotent cells are both positive for ΔPE-*Oct4*-GFP[4,23], the emergence of GFP⁺ /KLF4⁻ cells suggest the existence of a PGCLCs fate before reaching to naïve pluripotency during BiPNT (Supplementary Fig. 3a, b). We then quantified the putative PGCLC population with two documented PGC cell surface markers SSEA1 and CD61 by fluorescence-activated cell sorting (FACS) and show that 16.1% cells are SSEA1⁺/CD61⁺ at Day 6 BiPNT (Supplementary Fig. 3c).

To further investigate the PGCLCs emerged in BiPNT, we generated EpiSC cell line from ESC bearing *Blimp1*-mVenus (BV) and *Stella*-ECFP (SC) transgenes to monitor the PGCLCs fate[29]. Data from FACS and fluorescence microscope show that, *Prdm1*/*Blimp1*-mVenus is activated significantly when stimulated with Stage 1 medium, and reaches to a peak at 89% at Day 1, while the expression of *Stella*-ECFP is upregulated specifically when changed to 2iL medium, and the percentage of double positive for BVSC (BV⁺SC⁺) cells reaches to a peak at 13% at Day 6 (Fig. 3c and Supplementary Fig. 3d). In addition, the higher expression of *Prdm1*, *Prdm14*, *Tfap2c*, and *Dppa3* while the lower expression of naïve genes *Klf4* and *Tbx3* in Day 6 BV⁺SC⁺ cells distinguish them from naïve pluripotent cells (Fig. 3d). These data indicate an efficient Primed-PGCLC transition (PP$_G$T) before reaching to naïve pluripotency during BiPNT.

We further tested the GK15 or GK15 + cytokines (BMP4/SCF/EGF/LIF) culture conditions, which were used for PGC specification from ESCs previously[30,31], for their ability in PGCLCs induction at stage 2, and show by FACS that, BV⁺SC⁺ cells can be generated in both conditions (GK15 ~ 4.81%, GK15 + cytokines ~ 10.2%) (Supplementary Fig. 3e). These data suggest that the Day3 BiPNT cells have acquired the potential for germline specification.

To further capture the process of PGCLC-Naïve transition (P$_G$NT), we performed immunostaining for naïve pluripotent marker KLF4, and show that the sorted Day 6 BV⁺SC⁺ cells, when re-plated in 2iL medium, convert from KLF4⁻ cells to KLF4⁺ colonies gradually (Fig. 3e, f), suggesting a clear cell fate transition from PGCLCs to Naïve pluripotency.

Together, these results suggest a PGCLCs state along the NB branch during BiPNT.

**Characteristics of BV⁺SC⁺ PGCLCs.** To further characterize the Day6 BV⁺SC⁺ cells, we performed RNA-seq and compared our data to the published datasets which containing Day 4/Day 6 PGCLCs from ESCs in vitro or primary PGCs (E9.5,11.5,13.5) in vivo[32,33]. PCA analysis reveals that the transcriptomes of the

BV⁺SC⁺ cells are similar to D4 and D6 in vitro PGCLCs, and to a less extent to in vivo E9.5 PGCs (Fig. 4a). We then evaluated the epigenetic profiles of Day6 BV⁺SC⁺ cells. IF analysis reveals a reduced H3K9me2 and increased H3K27me3 in Day 6 BV⁺ cells (Fig. 4b), which are further confirmed by western blot (Fig. 4c), and consist with previous report[31]. We further determined the methylation states of maternally (*H19*) and paternally (*Snrpn*) imprinted genes in Day 6 BV⁺SC⁺ cells and show that whereas the DNA methylation in *Snrpn* loci is retained, the methylation states of *H19* loci reduced significantly, suggesting that Day 6 BV⁺SC⁺ cells may undergo a process of imprint erasure (Fig. 4d). Consistently, RT-qPCR analysis also shows the downregulation of de novo DNA methyltransferases *Dnmt3a/b* as well as *Uhrf1*, a critical regulator which recruits *Dnmt1* to hemi-methylated DNA (Supplementary Fig. 4a). Notably, BV⁺SC⁺ cells cultured in 2iL medium for one passage show a quick loss of *Blimp1* and can further form naïve-like colonies (Supplementary Fig. 4b). We further performed blastocysts microinjection experiments for the sorted BV⁺SC⁺ cells to determine their chimera formation ability (Supplementary Fig. 4c). Comparing with these naïve-like cells, Day 6 sorted BV⁺SC⁺ cells showed a poor chimera ability (12.5% versus 66.7%) (Supplementary Fig. 4c), consist with previous observations that PGCs cannot contribute to chimera, despite expressing naïve pluripotency genes[28,34,35].

To further determine the capacity of the Day 6 sorted BV⁺SC⁺ cells in spermatogenesis, we transplanted them into the seminiferous tubules of neonatal *W/Wᵛ* mice lacking endogenous germ cells[31,36,37]. Notably, 3 months after transplantation, tubules with normal spermatogenesis, which express the spermatogonia markers (DDX4 and PLZF), spermatocytes markers (SYCP3 and γH2AX), and spermatids markers (DDX4 and PNA) can be detected in two out of six testes sections transplanted with BV⁺SC⁺ cells, while only Sertoli cells in the tubules of control group (Fig. 4e, f and Supplementary Fig. 4d). These results confirm that Day 6 BV⁺SC⁺ cells are PGCLCs.

**Prdm1-KO blocks the generation of PGCLCs and naïve pluripotency.** To test whether PGCLCs are intermediates for the acquisition of naïve pluripotency, we knocked out *Prdm1*, a factor which is obligatory for PGC specification but not for pluripotency maintenance[38,39], in ΔPE-*Oct4*-GFP EpiSCs by CRISPR/Cas9 (Supplementary Fig. 5a–c). The *Prdm1⁻/⁻* EpiSCs exhibited indistinguishable morphology from wild-type (WT) EpiSCs and expressed comparable levels of primed pluripotency markers such as *Oct4*, *Sox2*, and *Otx2* (Fig. 5a and Supplementary Fig. 5d). However, when undergoing BiPNT, no GFP⁺ cells can be observed from *Prdm1⁻/⁻* cells, while the lentivirus-mediated overexpression of exogenous *Prdm1* can rescue this defect (Fig. 5b and Supplementary Fig. 5e, f). These data indicate that *Prdm1* is essential for the generation of PGCLC or naïve cells during BiPNT.

To further investigate the role of *Prdm1* in the cell trajectories during BiPNT, we performed single-cell analysis. To this end, we

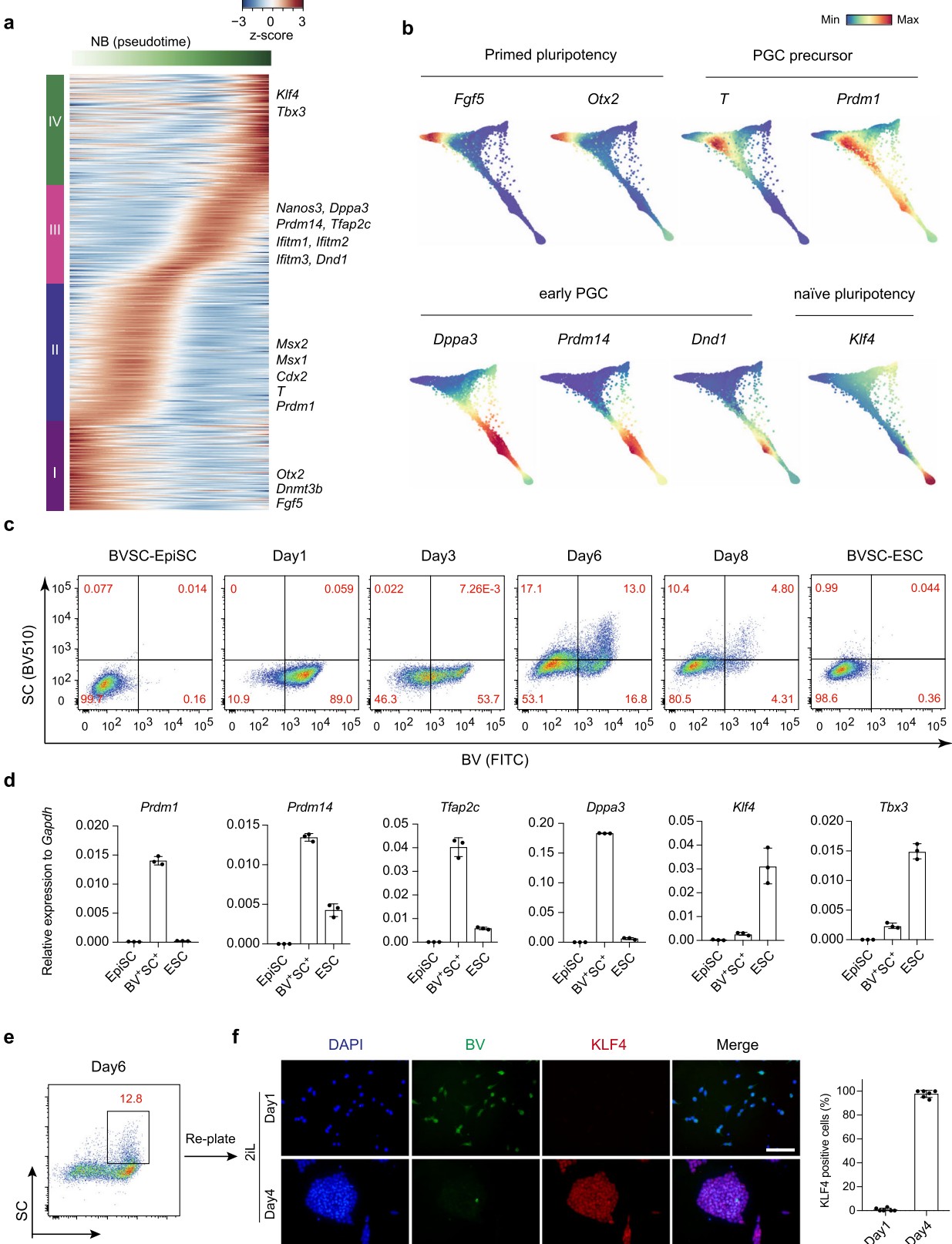

**Fig. 3 Activation of PGC-like program during successful BiPNT. a** Gene expression heatmap of 3219 dynamic expressed genes along NB (naïve branch) in a pseudotemporal order. Representative genes are shown on the right. **b** Plots showing the expression patterns of the indicated genes during BiPNT. **c** FACS analysis of BVSC induction at distinct time points during BiPNT. **d** RT-qPCR analysis for expression of indicated genes in EpiSC, BV⁺SC⁺ cells from Day 6 of BiPNT and ESC. Data are mean ± s.d., n = 3 independent experiments. **e** BV⁺SC⁺ cells were sorted from Day 6 of BiPNT as **c**. **f** Left: BV⁺SC⁺ cells from **e** were re-plated with 2iL medium for further transition into naïve state. Immunostaining for KLF4 to monitor the transition process. Scale bars, 100 μm. Right: Percentage of KLF4 positive cells. Data are mean ± s.d., n = 6 microscope fields. Source data are provided as a Source Data file.

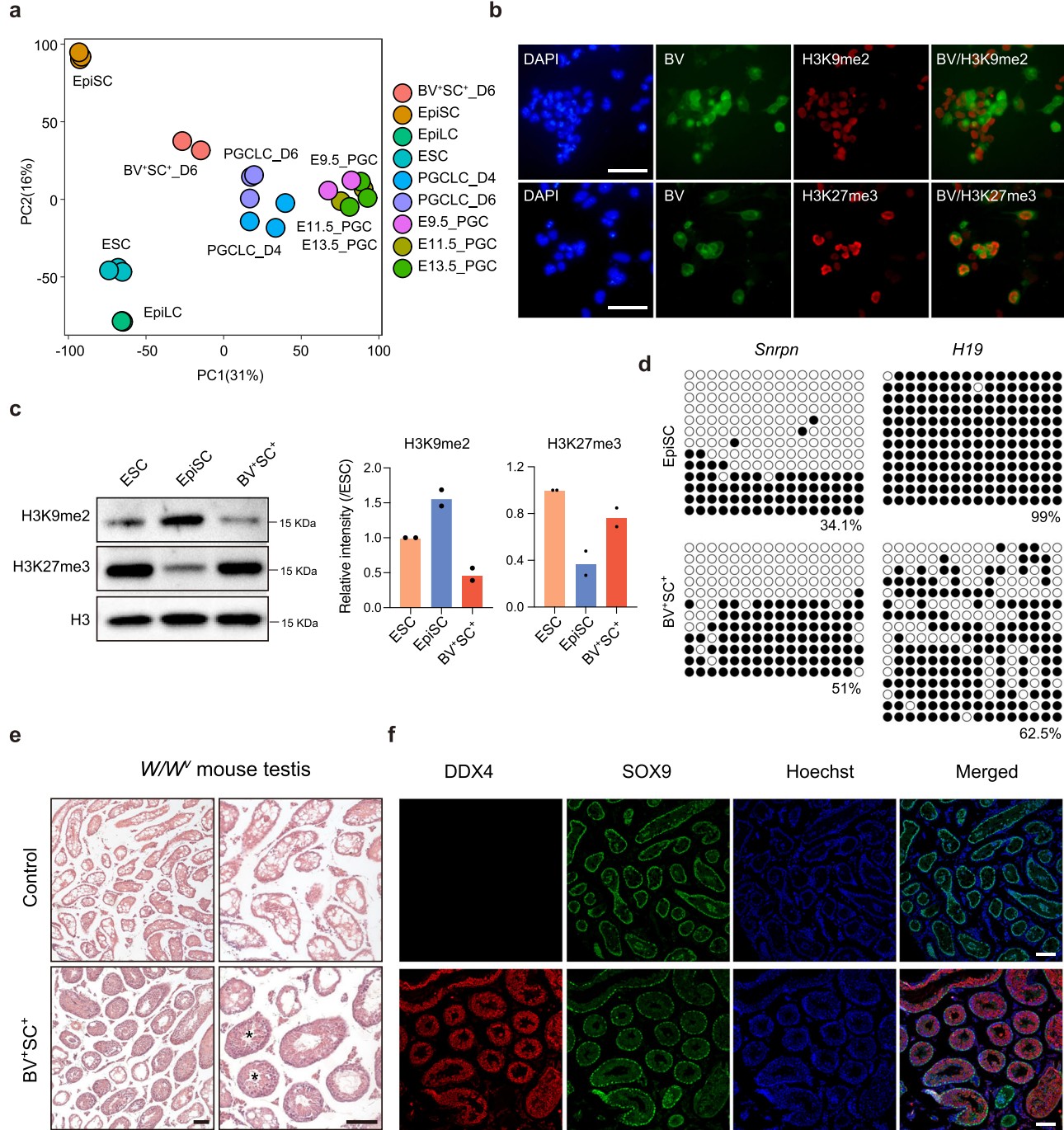

**Fig. 4 Characterization of PGC-like cells. a** PCA analysis of BV+SC+ cells at Day 6 of BiPNT, EpiSC, ESC, PGCLC-D4/6, E9.5 PGC, E11.5 PGC, E13.5 PGC, and EpiLC. **b** Immunofluorescence analysis of H3K9me2 and H3K27me3 in BV+ cells at Day 6 of BiPNT. Scale bars, 50 μm. **c** Western blot analysis (left) of H3K9me2 and H3K27me3 in ESC, EpiSC and BV+SC+ cells at Day 6 of BiPNT. Quantification of H3K9me2 and H3K27me3 normalized to H3 levels (right). Data are mean ± s.d., n = 2 independent experiments. **d** Bisulfite sequence analysis of 5mC of differentially methylated regions (DMRs) of the imprinted genes *Snrpn* and *H19* in EpiSC and BV+SC+ cells. White and black circles indicate unmethylated and methylated CpGs, respectively. **e** Testis sections from *W/W^v* mice and another one that was transplanted with Day 6 BV+SC+ cells stained by hematoxylin and eosin. Scale bars, 100 μm. **f** Immunofluorescence analysis of DDX4 and SOX9 expression in testis sections from *W/W^v* mice and another one that was transplanted with Day 6 BV+SC+ cells. Scale bars, 100 μm. The experiments in **b**, **e**-**f** were repeated twice with similar results. Source data are provided as a Source Data file.

collected scRNA-seq data for *Prdm1*⁻/⁻ EpiSCs undergoing BiPNT at corresponding time point same with WT EpiSCs (Fig. 5c). By comparing WT and *Prdm1*⁻/⁻ samples using UMAP, we show that the separation between WT and *Prdm1*⁻/⁻ populations occurs initially on Day1, and obviously on Day2 and Day3 (D2–3) (Fig. 5d). We further compared the expression pattern of selected genes between WT and *Prdm1*⁻/⁻ scRNA-seq

data, and show by heatmap that *Prdm1*-KO almost totally abolishes the activation of *Nanos3*, *Dppa3*, *Dppa5a*, and *Zfp42*, and also blocks, in varying degrees, the expression of *Snrpn*, *Ifitm1*, *Klf2*, and *Sox2*, while significantly increases the expression of the trophoblast genes, such as *Plac1*, *Igf2*, and *Ahnak* (Fig. 5e). RT-qPCR analysis further confirmed the expression pattern of these indicated genes (Fig. 5f).

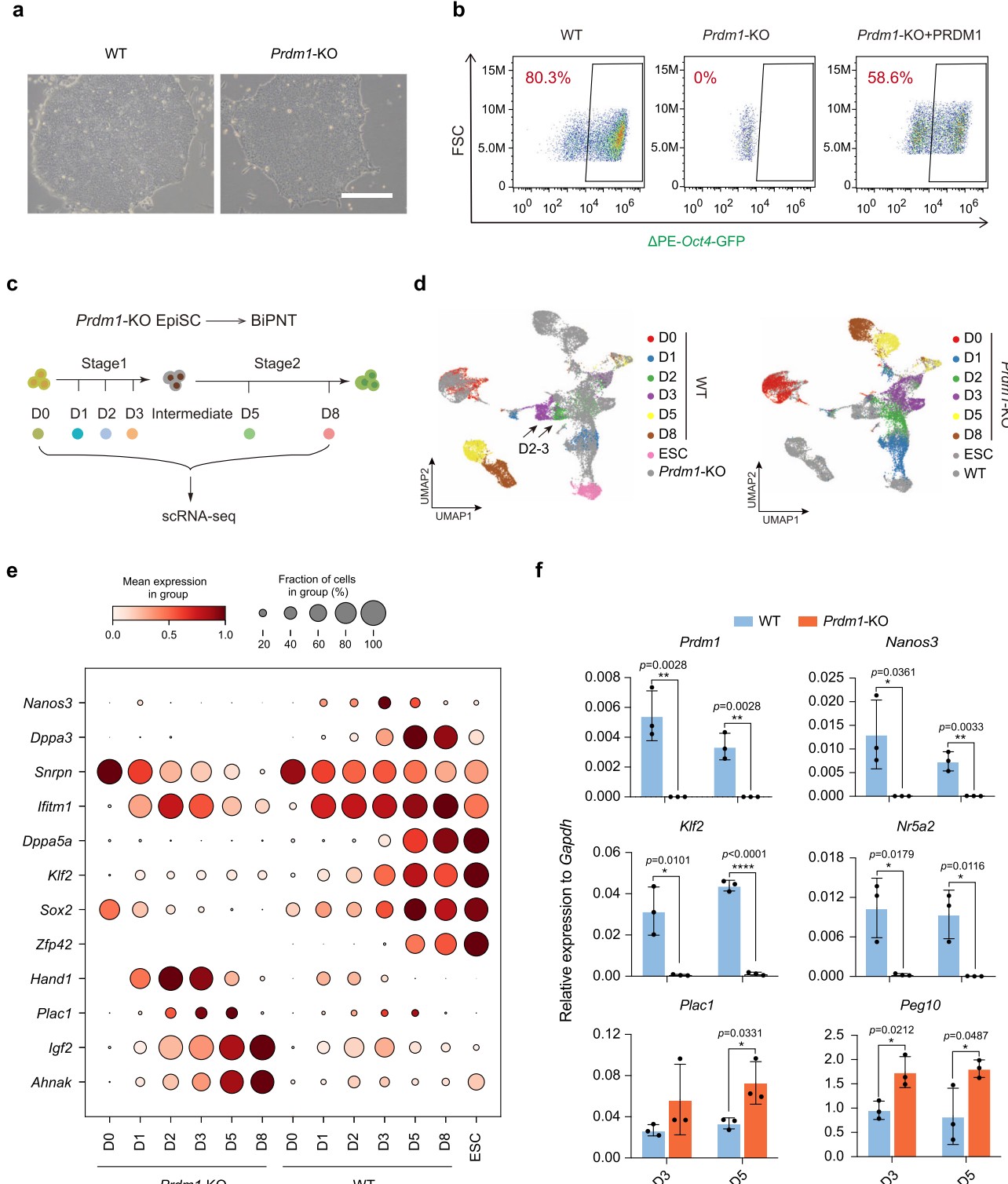

**Fig. 5 PGCLCs mediate the successful BiPNT. a** Representative images of wild-type (WT) and *Prdm1* knockout (*Prdm1*-KO) EpiSCs. Scale bars, 250 μm. Data shown are from 1 of 3 independent experiments. **b** FACS analysis of ΔPE-*Oct4*-GFP+ cells induced with WT, *Prdm1*-KO and rescued (overexpression of *Prdm1*) EpiSCs at Day 8. **c** Schematic diagram of scRNA-seq for *Prdm1*-KO EpiSC. **d** UMAP plots of WT and *Prdm1*-KO cells. WT cells are colored by the Day (D) during BiPNT while *Prdm1*-KO cells are marked with gray (left) or vice versa (right). The black arrow indicates the obviously depletion of D2-3 cells in *Prdm1*-KO EpiSC induced BiPNT. **e** Expression of indicated genes at the different time points. **f** RT-qPCR analysis for expression of indicated genes at Day 3 and Day 5 of WT or *Prdm1*-KO induced BiPNT. Data are mean ± s.d., two-tailed, unpaired student's *t*-test; *n* = 3 independent experiments. *$p < 0.05$, **$p < 0.01$, ****$p < 0.0001$. Source data are provided as a Source Data file.

The derived naïve cells in BiPNT show a distinctive methylation state from the EpiSCs and ESCs in loci of *Mest* (also known as *Peg1*), *Peg3* and *Snrpn* (Supplementary Fig. 5g), which may suggest their embryonic germ cell (EGC)-like feature. Furthermore, although the addition of JAK inhibitor in second stage of BiPNT can induce ΔPE-*Oct4*-GFP+ cells, they are almost negative for KLF4 (Supplementary Fig. 5h, i), indicating the block for the formation of naïve cells/EGC from PGCLCs, which is consistent with previous report that LIF/Stat3 signal is indispensable for PGC to EGC transition[40]. Together, these results suggest that *Prdm1*-dependent PGCLCs are the essential intermediates for naïve branch in BiPNT.

**DOT1Li restores PGCLC competence for primed pluripotency**. Developmentally, primed EpiSCs have lost the PGC competence[31]. The emergence of PGCLCs during BiPNT offers an ideal model in vitro to investigate the epigenetic barrier(s) established during PGC development in vivo. To this end, we performed dropout assay to compounds in stage 1 (S1) medium for identifying the critical factor(s) for PP$_G$T, and show that, the withdrawal of EPZ5676, a DOT1L inhibitor, blocks the induction of BV+SC+ PGCLCs significantly, while the addition of SGC0946, another specific DOT1L inhibitor, can rescue this defect, and seems even more potent than EPZ5676 (Fig. 6a). RT-qPCR analysis further validates the activation of PGC markers such as *Prdm1*, *Dppa3*, *Nanos3*, and *Prdm14* by DOT1L inhibitors (DOT1Li) (Fig. 6b and Supplementary Fig. 6a). Notably, DOT1L inhibition leads a significant repression for trophoblast markers such as *Gata2* or *Plac1* in late stage (Day 6) (Supplementary Fig. 6a), but not in early stage (Day 3) during BiPNT (Fig. 6b). These data indicate that DOT1L inhibition is critical for restoring PGCLC competence from EpiSCs and hinders trophoblast fate in late stage of BiPNT.

To gain any insight on how DOT1Li helps the restoring of PGCLC competence from EpiSCs, we performed RNA-seq in samples with or without DOT1Li (SGC0946) at Day3 BiPNT to probe the downstream targets and show that 330 genes, including the PGC markers *Prdm1* or PGC/Naïve pluripotency markers*Tfap2c*, *Nanos3*, *Nanog* and *Klf2* are upregulated, while 128 genes, including the HOX/GATA family members, such as *Hoxa*5/6/7/10, *Hoxb*2/3/4/6, *Hoxc*4/6/8/10/11, *Hoxd*9 and *Gata*3/6, are downregulated, by DOT1Li treatment, respectively (Fig. 6c). GO analysis shows that terms enriched in DOT1Li upregulated genes are involved in Wnt and BMP signaling pathway (Supplementary Fig. 6b), both of which are necessary for PGC specification as reported previously[27,41], while for terms enriched in DOT1Li downregulated genes are largely associated with lineage differentiation, such as embryonic skeletal system morphogenesis and limb morphogenesis (Supplementary Fig. 6b).

As chromatin accessibility dynamics (CAD) plays critical roles in gene expression regulation, we performed ATAC-seq, as done before[18], to investigate the effect of DOT1L inhibition on CAD at Day3 BiPNT, and show by heatmap that DOT1Li leads to chromatin OC (Open to Close) in 4831 loci (Fig. 6d), among which binding motif for TFs such as GATA1/2/3/4/6, HOXA9, HOXB4, HOXC9, HOXB13, and CDX2, could be enriched (Fig. 6e), while leads to chromatin CO (close to open) in 3251 loci(Fig. 6d), among which binding motif of PGC regulators or pluripotency factors such as NANOG, TFAP2C, OCT2/4/6 and ZIC1/3, could be enriched (Fig. 6e).

Based on the RNA-seq and ATAC-Seq data, we further tested a set of candidate transcriptional factors for their effect on BV+SC+ cells induction, and show that the BV+SC+ cell induction efficiency is promoted by overexpression of *Nanog* or *Tfap2c* separately (*Nanog*, 1.11–15.7%; *Tfap2c*, 1.11–11.2%) (Fig. 6f), and

can be further increased up to 22.8% by *Nanog* and *Tfap2c* co-expression when DOT1Li withdrawing (Fig. 6f and Supplementary Fig. 6c). In addition, when DOT1Li is present, the overexpression of *Nanog* or *Tfap2c*, separately or together, can further increase the BV+SV+ cells induction efficiency (Supplementary Fig. 6c). In contrast, the overexpression of *Gata3* or *Gata6*, two genes downregulated by DOT1Li, reduces the induction efficiency of BV+SV+ cells (Fig. 6f and Supplementary Fig. 6d). RT-qPCR analysis further confirmed the repression of PGC-related genes such as *Nanog*, *Prdm1*, *Dppa3* and *Nanos3* at Day3 BiPNT by the overexpression of *Gata3* or *Gata6* (Supplementary Fig. 6e).

DOT1L is a highly conserved histone methyltransferase that catalyzes H3K79 methylation[42], which is mainly found in active genes[43]. To gain insight into the relationship between H3K79 methylation and PGCLCs induction, we first performed western blot and show a significant decrease in global H3K79me2 at Day3 BiPNT upon DOT1Li (both EPZ5676 and SGC0946) treatment (Supplementary Fig. 6f). We then performed H3K79me2/H3K27me3 ChIP-seq for Day3 Cells with or without DOT1Li (SGC0946) treatment, and show that DOT1L inhibition leads a global loss of H3K79me2 (Fig. 6g). When analyzing the correlation between H3K79me2 and gene expression, we found that 30.6% upregulated genes, 58.0% downregulated genes, and 79.1% genes without change are marked by H3K79me2, respectively (Fig. 6h). Specifically, for genes such as *Gata3* and *Gata6*, the loss of H3K79me2 is associated with increased deposition of H3K27me3, and downregulation in mRNA (Fig. 6i), consistent with a role of H3K79me2 for active transcription[43]. For PGC genes such as *Nanog*, *Prdm1*, *Tfap2c*, and *Nanos3*, the loss of H3K79me2 is independent to H3K27me3, and associated with an upregulation in mRNA upon DOT1L inhibition (Fig. 6i). Since downregulated genes seem more sensitive to H3K79me2 (Fig. 6h), we further performed loss of function test for *Gata3/6* in BV+SV+ cells induction, and show that *Gata3* or *Gata6* knockout (KO) can facilitate the induction of BV+SV+ cells in the absence of DOT1Li (Fig.6j, k and Supplementary Fig. 6g–j). RT-qPCR further confirmed the upregulation of PGC-related genes such as *Nanog*, *Prdm1*, *Nanos3*, and *Dppa3* in *Gata3*- or *Gata6*-KO Cells (Supplementary Fig. 6k). In addition, DOT1Li can enhance the efficiency of BV+SC+ cell induction in *Gata3*- or *Gata6*-KO EpiSCs, comparing with WT EpiSCs (Fig. 6k).

Taken together, these data suggest that DOT1L inhibition promotes PP$_G$T in part by facilitating the loss of H3K79 methylation from *Gata3/6* and repressing their expression.

**Gata2 gates trophoblast-like whereas hampers PGCLC program**. To further determine the key events for early cell fate choice during BiPNT, we reanalyzed the scRNA-seq data at D0, D1, and D2 by UMAP. We found a significant cell diversity at D2 (Fig. 7a). Heatmap for representative genes show an activation of trophoblast signal (*Gata2*, *Plac1*) as early as Day 1 (Fig. 7b), and a diversity of pluripotent signal (*Dppa3*, *Nanog*) at Day 2 (Fig. 7b), suggesting an early cell fate choice. To further investigate the key factor(s) regulate this cell fate choice, we use pySCENIC[44] to predict the regulon of each cell subgroup and heatmap illustrated a widely activation of early BMP4 response regulons, such as *Id1*, *Cdx2*, *Tfap2c*, and a separate pattern for regulons *Prdm1* (PGC) and *Gata2* (Trophoblast) at D2 populations (Fig. 7c and Supplementary Fig. 7a). The *Prdm1*-KO almost abolished PNT, indicating *Prdm1* as a key regulon for PGC-Naïve trajectory. To test whether *Gata2* is a regulon specific for trophoblast trajectory, we generated two independent *Gata2*-KO EpiSC cell lines in BVSC EpiSCs by CRISPR/Cas9 (Supplementary Fig. 7b, c). These two cell lines express comparable levels of primed

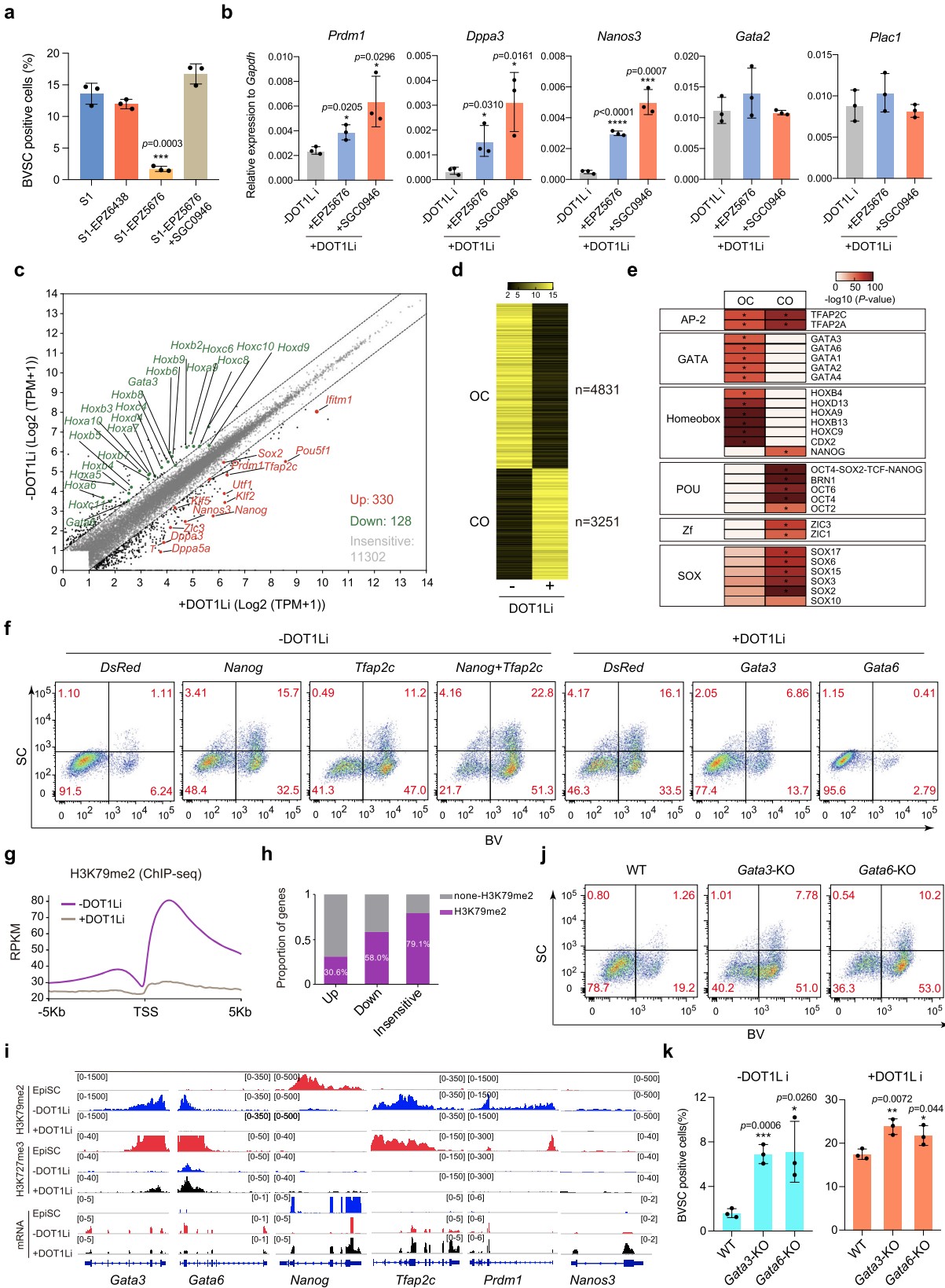

pluripotency markers (Supplementary Fig. 7d). Comparing with WT, *Gata2*-KO shows a significant increase of BV⁺ cells at Day3 and BV⁺SC⁺ cells at Day6 during BiPNT (Fig. 7d). Consistently, RT-qPCR analysis shows almost abolishment in trophoblast signal such as *Plac1*, *Peg10*, and *Phlda2*, and a significant increase in PGC-like signal such as *Prdm1*, *Prdm14*, *Dppa3*, and *Nanos3* in

*Gata2*-KO cells during BiPNT (Fig. 7e). We further sorted *c-Kit* for WT and two *Gata2*-KO cell lines and show that *Gata2*-KO leads to a significant increase of *c-Kit*⁺ population at Day3 BiPNT (WT ~ 53%, KO1# ~ 76.2%, KO2# ~ 75.4%) (Supplementary Fig. 7e). RT-qPCR analysis further confirmed the loss of trophoblast markers in *Gata2*-KO *c-Kit*⁻ cells (Supplementary

**Fig. 6 DOT1L inhibitor is responsible for the induction of PGCLCs from EpiSCs. a** FACS analysis of BVSC induction at Day 6 under different conditions. S1, stage1. **b** RT-qPCR analysis for expression of PGC and trophoblast markers at Day 3 of BiPNT without (−) or with (+) DOT1L inhibitor (DOT1Li: EPZ5676 and SGC0946) treatment. Data are mean ± s.d., two-tailed, unpaired student's *t*-test; $n = 3$ independent experiments. *$p < 0.05$, ***$p < 0.001$, ****$p < 0.0001$. **c** The difference between the trans**c**riptomic profiles without (−) or with (+) DOT1Li (SGC0946) treatment at Day 3. Representative significantly upregulated genes (compared with −DOT1Li, log2 fold change (log2 FC > |1|)) are in red, and downregulated ones are in green. **d** Heatmap of open or closed chromatin loci in Day 3 cells without (−) or with (+) DOT1Li (SGC0946) treatment. CO: Closed to Open, OC: Open to Closed. **e** Motif discovery for DOT1Li (SGC0946) mediated CO and OC loci. **f** FACS analysis of BV+SC+ PGCLCs induction at Day 6 after overexpression of *DsRed* (control), *Nanog*, *Tfap2c* or combined without (−) DOT1Li (SGC0946) treatment, or overexpression of *Gata3* and *Gata6* with (+) DOT1Li (SGC0946) treatment. **g** Metaplot of H3K79me2 reference-normalized ChIP-seq signal at all transcriptional start site (TSS) in Day 3 BiPNT cells without (−) or with (+) DOT1Li (SGC0946) treatment. **h** Proportion of differentially expressed genes and insensitive genes from **c**, which are directly marked with H3K79me2 in Day 3 cells (-DOT1Li). **i** ChIP-seq (H3K79me2, H3K27me3) and RNA-seq tracks of indicated genes in untreated EpiSC and Day 3 BiPNT cells without (−) or with (+) DOT1Li (SGC0946) treatment. **j** FACS analysis of BVSC induction at Day 6 with *Gata3*- or *Gata6*-KO EpiSCs. **k** Percentage of BVSC positive cells induced with *Gata3*- or *Gata6*-KO EpiSCs under different conditions. Data are mean ± s.d., two-tailed, unpaired student's *t*-test; $n = 3$ independent experiments. *$p < 0.05$, **$p < 0.01$, ***$p < 0.001$. Source data are provided as a Source Data file.

Fig. 7f). These data suggest that *Gata2*-KO results in an early alteration of lineage trajectories during BiPNT, which turns off the trophoblast-like program while facilitates the PGC-like program.

## Discussion

In this report, by performing scRNA-seq to a previous well-established PNT process driven by BMP4 (BiPNT)[18], we generated a cell fate continuum between primed and naïve pluripotency states and show that the trajectories of BiPNT can bifurcate into *c-Kit*+ naïve and *c-Kit*− trophoblast-like branches. Surprisingly, the naïve branch, as shown here (Fig. 8), in single-cell level, goes through a PGC-like intermediate, which links PNT with two distinctive while continuous cell fate transitions, here they were termed as Primed to PGCLC transition or PP$_G$T, and PGCLC to Naïve transition or P$_G$NT, respectively. Our finding shows a unique landscape for resetting the stable mouse primed pluripotent state into a naïve pluripotent one by linking an important developmental intermediate state therefore sheds light on the mechanism of cell reprogramming and differentiation.

*Prdm1* and PGC-like cells have been demonstrated to be dispensable for the derivation and maintenance of ESCs as well as the reprogramming of EpiSCs into rESCs[39,45]. The obligatory role of *Prdm1* in our study may be due to the differences of induction systems, e.g., BMP4, a key growth factor known for driving PGC/PGCLC specification. In addition, we previously showed that BMP4 is able to activate naïve or PGC-related genes such as *Klf2*, *Esrrb*, *Tfap2c*, and *Dppa3*[18], which further underlines the PGCLC intermediates for recapturing naïve pluripotency in BiPNT. Actually, the BV+SC+ cells induced during BiPNT exhibited PGC features, including upregulation of early PGC markers, increase of H3K27me3 and reduction of H3K9me2, erasure of imprinting, poor contribution to blastocyst chimeras and contribution to spermatogenesis, indicating their PGCLCs fate. It's of great interest to test the function of the sperm/spermatids generated from those BV+SC+ cells in future.

In mice, permissiveness for PGC specification is unique to formative epiblast cells (i.e., E5.0–E6.0 epiblast, EpiLCs) or PSCs with formative pluripotency[31,41,46,47]. The primed EpiSCs, which corresponding to E6.5 epiblast, have lost the competence of PGC responsiveness[31,41]. Nevertheless, in BiPNT system, a PGC-like program is robustly activated from the primed EpiSCs. The critical role of DOT1Li identified here indicates H3K79 methylation may act as an epigenetic barrier for PP$_G$T. Mechanistically, DOT1Li promotes the generation of PGCLC partially through removing H3K79me2 from lineage genes such as *Gata3* and *Gata6*, leading to their repression (Fig. 8). Interestingly, DOT1Li mediated loss of H3K79me2 is also observed at chromatin loci of upregulated genes, such as PGC genes *Tfap2c*, *Prdm1*, *Nanos3*,

etc. Indeed, H3K79 methylation has been linked to transcriptional repression in some cases[48–51], suggesting a context dependent manner for H3K79 methylation on transcriptional regulation. However, as a secondary effect of DOT1Li on activation of PGC genes may exist, future study is needed to investigate whether H3K79me2 also play a repressive role in BiPNT as well as the underlying mechanism.

## Methods

**Mice**. 129 Sv/Jae and ICR mice were purchased from Beijing Vital River Laboratory, and ΔPE-*Oct4*-GFP (OG2) transgenic allele-carrying mice (CBA/CaJ X C57BL/6 J) were purchased from The Jackson Laboratory. *W/W*^v mice were provided by Dr. Xiaoyang Zhao. All animal experiments were operated under the Animal Protection Guidelines of Guangzhou Institutes of Biomedicine and Health, Chinese Academy of Sciences, Guangzhou, China.

**EpiSCs derivation and cell culture**. Mouse EpiSCs were derived from E5.5 mouse embryos by crossing male homozygous ΔPE-*Oct4*-GFP transgenic allele-carrying (CBA/CaJ X C57BL/6 J) with 129 Sv/Jae female mice, following the described protocols[3,52]. After culturing on feeder for the first 4–8 passages, stable EpiSCs were maintained feeder-free on fetal bovine serum (FBS)-coated dishes in N2B27 + bFGF (15 ng/ml, PeproTech) + Activin A (20 ng/ml, PeproTech) + XAV939 (1 μM, Selleck) (FAX) medium. N2B27 medium comprised equal volume of DMEM/F12 (GIBCO) and Neurobasal (GIBCO), 0.5% N2 (GIBCO), 1% B27 (GIBCO), 1% GlutaMAX (GIBCO), 1% non-essential amino acids (NEAA) (GIBCO) and 0.1 mM β-mercaptoethanol (GIBCO). EpiSCs were dissociated with Accutase (Sigma) and passaged as singular cell with FAX medium in the presence of 5 μM Y27632 (Selleck) at ~20,000 cells in a well of 12-well plate every 3 days. The medium was refreshed daily.

Mouse ESCs were maintained feeder-free on 0.1% gelatin-coated dishes in N2B27 medium supplemented with 2i/LIF medium as described[18].

**Derivation of EpiSCs from mouse BVSC-ESCs**. Mouse BVSC-ESCs were dissociated into single cells using 0.05% Trypsin-EDTA and were plated at a density of 2–5 × 10^4 in a well of 12-well plate coated with FBS in N2B27-2iL medium. The next day, medium was changed to FAX medium for 3 days and then re-plated at a ratio of 1:10. Cells were passaged with Accutase every 3 days. Experiments were performed between P8 and P20.

**BMP4 induction of EpiSCs into naïve state**. Reprogramming of EpiSCs into naïve state was performed with our protocol published previously with minor modification[18].

**Reagents setup**. Stage 1 medium: iCD1 medium[53] comprised Vitamin C (50 μg/ml, Sigma), bFGF (10 ng/ml), LiCl (5 mM, Sigma), LIF (1000 U/ml, Millipore) and CHIR99021 (1 μM, synthesized in GIBH), with the addition of BMP4 (5–10 ng/ml, R&D Systems), EPZ6438 (1 μM, TargetMol) and EPZ5676 (2.5 μM, TargetMol) or SGC0946 (1 μM, TargetMol). In some experiments, we also added 2 μM Parnate (Selleck).

Stage 2 medium: N2B27-2iL medium comprised N2B27 medium with the supplement of 50 μg/ml Vitamin C, 1 μM PD0325901, 3 μM CHIR99021 and 1000 U/ml LIF.

2. EpiSCs were passaged with Accutase into singular cell and seeded at a density of 5,000 cells per well of 24-well plate coated with FBS in N2B27 + FAX medium with the addition of 5 μM Y27632. Next day, the medium was replaced with Stage 1

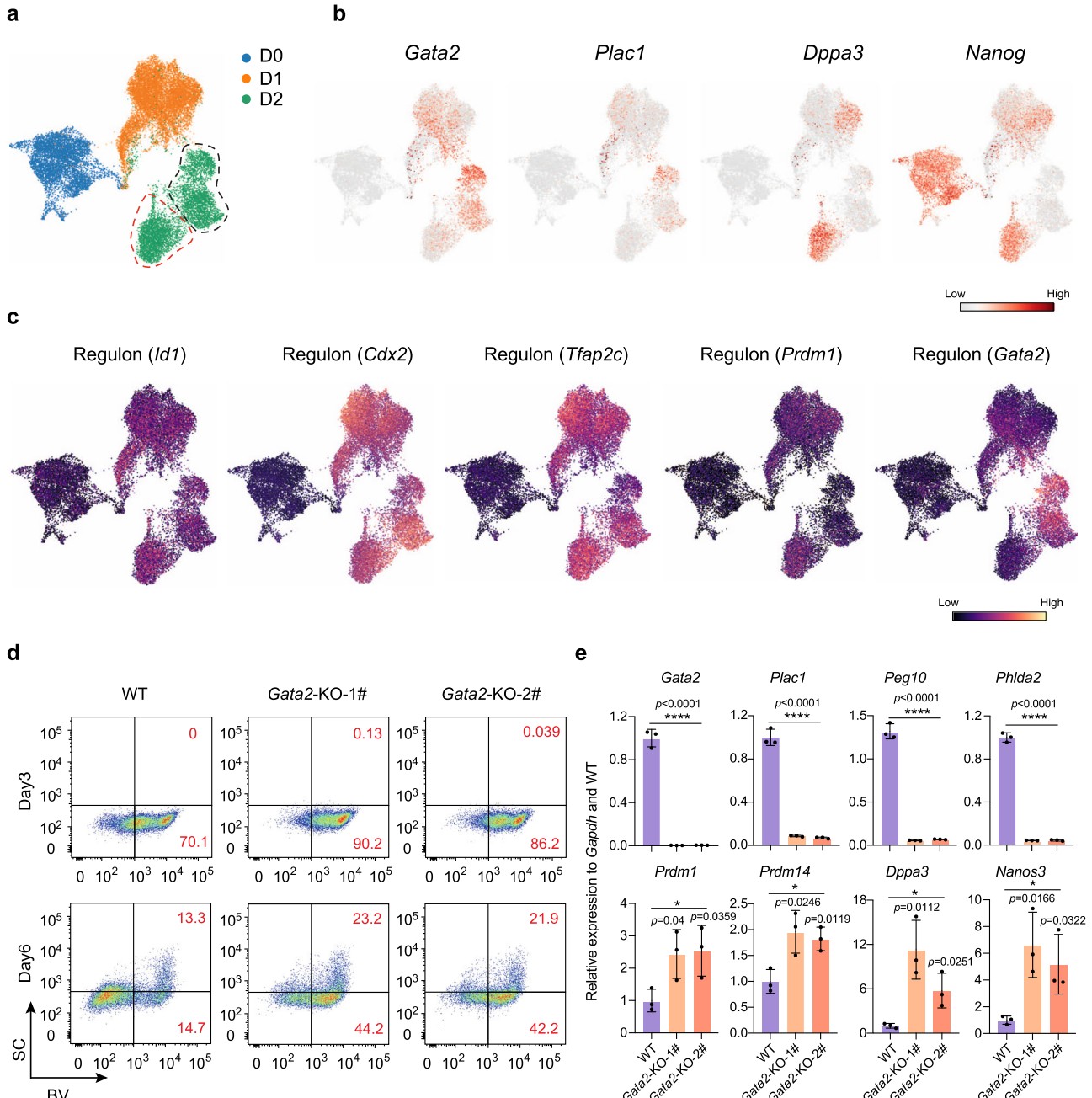

**Fig. 7 Deficiency of Gata2 hinders trophoblast fate but promotes PGCLCs. a** t-SNE representation of 24233 cells during the first three days of whole process, colored by time point of samples. The cells enclosed by red and black dotted lines indicate two distinct populations. **b** Cells colored by expression pattern of *Gata2*, *Plac1*, *Dppa3*, and *Nanog* in t-SNE representation. **c** t-SNE plot displaying *Id1*, *Cdx2*, *Tfap2c*, *Prdm1* and *Gata2* regulons (red dots, active; black dots, inactive) in the first 3 days, respectively. **d** FACS analysis of BVSC induction at Day 3 and Day 6 during BiPNT in WT or two independent *Gata2*-KO EpiSCs. **e** RT-qPCR analysis for expression of trophoblast and PGC associated genes in WT and two *Gata2*-KO cells from Day 6 of BiPNT. Data are mean ± s.d., two-tailed, unpaired student's *t*-test; $n = 3$ independent experiments. *$p < 0.05$, ****$p < 0.0001$. $n = 3$ independent experiments. Source data are provided as a Source Data file.

medium and changed daily for the first 3 days. The Stage 2 medium was refreshed daily for the last 5 days.

**Flow cytometry**. Cells were dissociated with 0.05% Trypsin-EDTA and neutralized with DMEM containing 10% FBS. After the dissociation, cells were collected by centrifugation and resuspended with flow cytometry buffer (DPBS with 0.1% BSA and 1 mM EDTA) at a density of $1 \times 10^6$ cells per 100 μl buffer. Cells were then incubated with fluorophore-conjugated antibodies for 30 min at 4 °C. After incubation, cells were washed with DPBS for three times, resuspended with flow cytometry buffer, and filtered with 40 μm cell strainer to remove large clumps of cells. Cells were then analyzed with LSR Fortessa X-20 (BD Biosciences) or sorted

with BD FACSAria III (BD Biosciences). The FACS data were analyzed with FlowJo 10.4 software.

The following antibodies were used: APC anti-c-Kit (eBioscience, 17-1171-83,1:200), Alexa Fluor 647 anti-mouse/human CD15 (SSEA-1) (BioLegend, 125607, 1:200), PE anti-mouse/rat CD61 (BioLegend, 104307, 1:500).

**Immunofluorescence**. The cells were seeded on coverslips and the fixed with 4% paraformaldehyde for 30 min at room temperature. The cells were washed with PBS once and subsequently soaked in the mixing buffer of equal volume of 0.1% Triton X-100 and 3% BSA for 1 h at room temperature. The cells were washed with PBS once, then incubated with primary antibody at 4 °C for overnight. The cells were washed with PBS five times next day, then incubated with appropriate

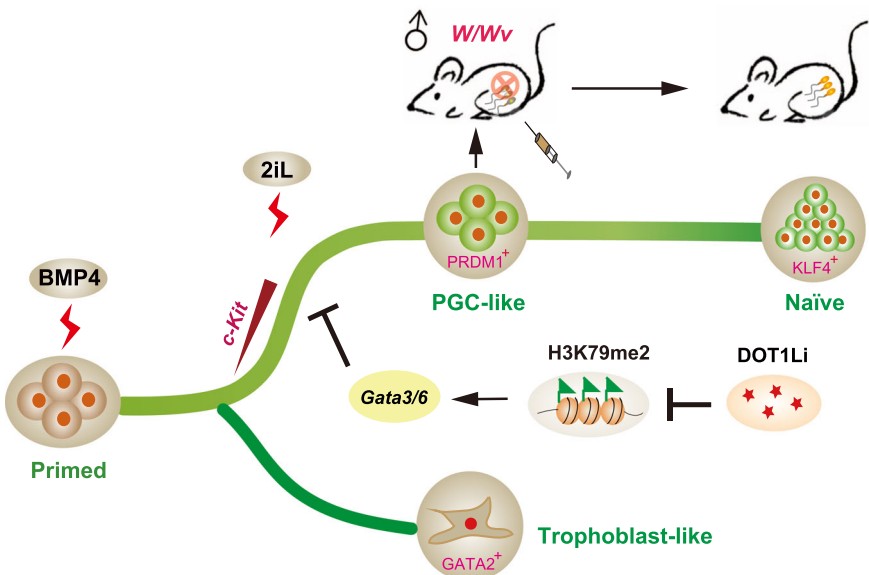

**Fig. 8 Schematic diagram of cell fate trajectories during BiPNT.** During BiPNT, EpiSCs bifurcate into *c-Kit*+ naïve and *c-Kit*- trophoblast-like cells. A PGC-like intermediate which capable of spermatogenesis in vivo is emerged before reaching to the naïve pluripotent state. The inhibition of DOT1L activity permits the transition from primed pluripotency to PGCLCs in part by facilitating the loss of H3K79me2 from *Gata3/6* and repressing their transcription.

secondary antibody for 1 h at room temperature. The cells were washed with PBS five times to remove the secondary antibody and counterstained with DAPI for 3 min and washed with PBS once. In the end, the coverslips were mounted on the slide for observation under the confocal microscope (Zeiss 710 NLO). Primary antibodies and secondary antibodies were diluted in 3% BSA. Primary antibodies used in this study were anti-KLF4 (R&D system, AF3158, 1:100), anti-H3K9me2 (Abcam, ab1220, 1:100), anti-H3K27me3 (Merck, 17–622, 1:200), anti-PRDM1 (Invitrogen, 14–5963–82, 1:50), anti-DDX4 (Abcam, ab13840, 1:400), anti-SOX9 (EMD Millipore, AB5535, 1:400), anti-PLZF (Santa Cruz, sc-28319, 1:100), anti-SYCP3 (Abcam, ab15093, 1:400), and anti-γH2AX (Abcam, ab26350, 1:400), anti-PNA (ThermoFisher, L21409, 10 μg/ml).

**Western blot.** $1 \times 10^6$ cells were harvested and lysed with 100 μL lysis buffer which containing 62.5 mM Tris-HCl (pH 6.8), 2% SDS, 10% glycerol, 0.025% bromophenol blue and 50 mM DTT with protease inhibitor cocktail (Roche) on ice for 5 min and boiled for 10 min at 100 °C. The samples were separated by 10–12.5% SDS-PAGE and transferred to a PVDF membrane (Millipore) using a wet transfer system, and then incubated with the primary antibodies and secondary antibodies. The following antibodies were used: anti-H3K9me2 (Abcam, ab1220,1:500), anti-H3K27me3 (Merck,17–622,1:2000), anti-H3 (Abcam, ab1791, 1:2000), anti-H3K79me2 (Active Motif, 39143, 1:1000), anti-PRDM1 (Invitrogen, 14–5963–82, 1:100) and anti-ACTIN (Sigma, A5441, 1:4000).

**Transplantation of the BV+SC+ cells into Seminiferous Tubules of Neonatal *W/W*ᵛ Mice.** Recipient animals (neonatal [10- to 15-day-old] *W/W*ᵛ male mice lacking endogenous spermatogenesis) were induced into hypothermic anesthesia on ice. The glass needle with a 40 μm inner diameter was filled with $2 \times 10^5$ cell/μL sorted BV + SC + cells and the donor cell suspension (~2 μl) was injected into the efferent duct of recipient *W/W*ᵛ mice testis. Then the recipient animals were returned to their littermates after surgery and they were sacrificed by anesthesia at 2 months after transplantation and testicular tissue were fixed in 4% paraformaldehyde overnight at 4 °C for Immunofluorescence staining.

**Quantitative RT-PCR.** Total RNA was extracted using the TRIzol. cDNA was reversed transcribed by HiScript II Reverse Transcriptase (Vazyme biotech R222-01) from 2 μg of total RNA. Real-time quantitative PCR was performed with ChamQTM SYBR® qPCR Master Mix (Vazyme biotech Q311-02). Relative expression values were normalized to *Gapdh* and analyzed with Microsoft Excel 2019. Each experiment was operated in technical triplicate. Primers for RT-qPCR were listed in Supplementary Data 1.

**Gene editing in EpiSCs.** Knockout EpiSCs lines were obtained by the CRISPR/Cas9 system. Guide RNAs (gRNAs) were designed in website: crispr.mit.edu., and cloned downstream of human U6 promoter of pX330 and pXP (modified from pX330, with the resistance of puromycin). For *Prdm1, Gata2, Gata3* and *Gata6* knockout, two pairs of gRNAs were designed to delete the critical exons. For establishing *Gata2*-tdTomato reporter, two pairs of gRNAs were used to make cleavages and a tdTomato cassette was targeted into exon1 of *Gata2* by

homologous recombination. The gRNAs and linearized donor plasmids were transfected with Lipofectamine 3000 (Thermo Fisher Scientific) in EpiSCs followed the guidance of manufacturer. $1 \times 10^5$ cells were resuspended in FAX medium with Y27632 in one plate of the 12-well plate. After 8–10 h, the medium was changed with fresh FAX medium. Forty-eight hour post transfection, the cells were selected with puromycin for 3 days and 1000 cells were seeded on one well of the 6-well plate for colony formation. Singular colony were picked up and evaluated by genotyping. Primers for gRNAs were listed in Supplementary Data 2.

**Bisulfite sequencing.** The genomic purification kit (Promega) was used to isolate genomic DNAs, and EpiTect Bisulfite Kit (QIAGEN) was used to perform the bisulfite reactions. LA Taq (TaKaRa) was used to amplify differentially methylated regions of *Snrpn, H19, Peg1*, and *Peg3*. The PCR products were subcloned into the pMD-18T vector (TaKaRa), and then sequenced. The results were analyzed in quma.cdb.riken.jp. Primers were listed in Supplementary Data 3.

**RNA-seq and data analysis.** Total RNA was isolated. Sequencing libraries were prepared according to the VAHTSTM mRNA-seq V3 Library Prep Kit for Illumina® (NR611-01/02, Vazyme) and sequenced using a NextSeq 500 High Output Kit V2 (75 cycles) (FC-404–1005, Illumina).

RNA-seq was proceeded as previously described[18]. Briefly, sequenced reads were aligned to the GENCODE annotations transcriptome (M13) using Bowtie2 (version 2.2.5)[54] and RSEM (version 2.4.1)[55], and normalized using EDASeq (version 2.4.1)[56]. Differentially expressed genes were obtained using DESeq2 (version 1.22.1)[57] with *P*-value < 0.05 and fold change > 2 as a threshold. Gene ontology analysis was performed using goseq (version 1.22.0). PCA was performed using function prcomp.

**ATAC-seq.** ATAC-seq was executed as previously described[18]. Briefly, a total of 50,000 cells were washed with 50 μL cold PBS once, then centrifuged for 5 min at 500 g at 4 °C, then resuspended in 50 μL lysis buffer (10 mM Tris-HCl pH 7.4, 10 mM NaCl, 3 mM MgCl₂, 0.2% (v/v) IGEPAL CA-630) and incubated at 4 °C for 10 min. The suspension of nuclei was centrifuged for 5 min at 500 g at 4 °C, followed by the addition of 50 μL transcription reaction mix (10 μL 5X TTBL, 5 μL TTE Mix V50, and 35 μL nuclease-free H₂O) of TruePrep TM DNA Library Prep Kit V2 for Illumina® (TD501–503, Vazyme biotech) and incubated at 37 °C for 30 min. DNA fragments were purified by MinElute Kit (QIAGEN). ATAC-seq libraries were amplified by 13 cycles of amplification according to the TruePrep TM DNA Library Prep Kit V2 for Illumina® (TD501–503, Vazyme biotech), then the libraries were purified by a Qiaquick PCR (QIAGEN) column. Library concentration was detected by a VAHTS TM Human Genomic DNA Quantification and QC Kit (NQ201, Vazyme biotech) according to the manufacturer's instructions. The concentration of ATAC library peaks was measured with Real-Time Quantitative PCR by detecting the differential expression of housekeeping gene (GAPDH) and negative genes. In the end, the ATAC library was sequenced on a NextSeq 500 using a NextSeq 500 High Output Kit V2 (150 cycles) (FC-404–2002, Illumina) according to the manufacturer's instructions.

**ChIP-seq and data analysis**. ChIP assay was carried out as previously described[58]. The cells were crosslinked with 1% formaldehyde for 10 min and then quenched by adding 125 mM glycine. Chromatin samples were lysed with lysis buffer (20 mM Tris-HCl, pH 8.0, 500 mM NaCl, 1 mM EDTA, 1% Triton X-100 and 0.1% SDS) and sonicated with Qsonica. Histone specific antibody was incubated with chromatin samples overnight at 4 °C. Antibodies used were as follows: H3K79me2 (Active motif, 39143), H3K27me3 (CST, 9733 S). About $5 \times 10^5$ cells were used for each ChIP. We used 0.5 µg spike-in antibody (Active motif, 61686) and 25 ng spike-in chromatin (Active motif, 53083) in this ChIP assay according to the manufacturer's guidelines. The protein-DNA complexes were immobilized on 15 µl protein A/G beads (Smart lifesciences, SA032005) and then washed three times with lysis buffer, twice with low salt buffer (10 mM Tris-HCl, 250 mM LiCl, 1 mM EDTA, 0.5% NP40, 0.5% Na-deoxylcholate) and once with TE buffer (10 mM Tris-HCl, pH 8.0). Decrosslinking was carried out in elution buffer (50 mM Tris-HCl, pH8.0, 10 mM EDTA, and 1% SDS) at 65 °C for at least 5 h. Proteinase K and RNase A digestions were performed at 55 °C for 1 h. DNA samples were purified with PCR extraction kit (QIAGEN, 28006). DNA samples were analyzed using real-time PCR and prepared for deep sequencing according to the manufacturer's guidelines (Vazyme, ND607). Finally, libraries were pooled and sequenced on the Illumina NovaSeq 6000 sequencer with paired-end sequencing.

ChIP-seq data analysis were performed as previously described[18]. In brief, Sequenced reads were aligned to the mouse reference genome (mm10) using Bowtie2[54] with default parameters. Reads that mapped to mitochondrial DNA or unassigned sequences were discarded. For paired-end sequencing data, only concordantly aligned pairs were retained. Peaks were called using MACS2 (version 2.1.2)[59] with the default parameters. Alignment BAM files were transformed into read coverage files (bigWig format) using deepTools (version 2.5.4)[60] using the RPKM normalization method.

**Single-cell sequencing**. Cells at different induction time points of BiPNT were collected and resuspended in DPBS with 0.04% BSA. Then, cells suspensions (500–1000 cells per microliter) were loaded on a Chromium Single Cell Controller (10x Genomics) to obtain single-cell gel beads in emulsion (GEMs) by using Single Cell 3' Library and Gel Bead Kit V2 (10x genomics, 120237). Captured cDNAs were lysed and the released RNA were barcoded through reverse transcription in singular GEMs. Barcoded cDNAs were pooled and cleaned by DynaBeads® MyOne Silane Beads (Invitrogen, 37002D). Single-cell RNA-seq libraries were prepared by Single Cell 3' Library Gel Bead Kit V2 (10x Genomics, 120237) following the manufacturer's instruction. Sequencing was operated on an Illumina HiSeq X Ten with pair end 150 bp (PE150).

**Single-cell RNA-seq data analysis**. Fastq reads were aligned to the genome using STARsolo[61] with the setting '--outSAMattributes NH HI AS nM CR CY UR UY --readFilesCommand zcat --outFilterMultimapNmax 100 --winAnchorMultimapNmax 100 --outMultimapperOrder Random --runRNGseed 777 --outSAMmultNmax 1'. The count matrix was lightly filtered to exclude cell barcodes with low numbers of counts: Cells with <1000 UMIs, <500 genes or >20% fraction of mitochondrial counts were removed. The filtered matrix was normalized using scran[62]. The top 4000 most highly variable genes were used for PCA, and the first 50 PCs (principle components) were used for downstream analysis with SCANPY[63]. Single-cell trajectory was analyzed by Harmony[25] and the top 1000 highly variable genes were used for PCA, and the force-directed layout was computed using first 150 PCs. The genes expression trajectories on pseudotemporal orderings of cells (Fig. 3a) were analyzed by LineagePulse (https://github.com/YosefLab/LineagePulse) according to the pseudotime generate from Harmony. The *Prmd1* KO and WT scRNA-seq data batch effect was corrected by Seurat[64].

**Statistics and reproducibility**. No statistical method was used to predetermine the sample size. The experiments were not randomized. The investigators were not blinded to allocation during the experiment and outcome assessment. Data are presented as mean ± s.d. *P*-values were calculated using two-tailed unpaired Student's *t*-tests with GraphPad Prism 6. $P < 0.05$ was considered to be statistically significant; *$P < 0.05$, **$P < 0.01$, ***$P < 0.001$. The exact *P*-values are indicated in relevant figures. The statistical test, numbers for sample sizes, and independent experiments are indicated in the figure legends.

**Reporting summary**. Further information on research design is available in the Nature Research Reporting Summary linked to this article.

## Data availability

The scRNA-seq, RNA-seq, ATAC-seq, and ChIP-seq data generated in this study have been deposited in the Gene Expression Omnibus (GEO) database under accession code GSE147088 Source data are provided with this paper. The source data underlying Figs. 2j, 3d, f, 4c, 5f, 6a, b, k, 7e and Supplementary Figs. 3b, 4a, 5d, i, 6a, c–e, k, 7d, f are provided as a Source Data file. The authors declare that all data supporting the findings of this study are available within the article and its supplementary information files or from the corresponding author upon reasonable request.

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

## Acknowledgements

We thank Dr. Mitinori Saitou for BVSC ESC cell lines. This work was supported by grants from National Natural Science Foundation of China (U20A2013 [C.Z.], 31970681 [J.L.], 32022019 [J.L.], 32100594 [C.Z.], 31421004 [D.P.], 31530038 [D.P.], 21907095 [S.C.]), The National Key Research and Development Program of China, (2016YFA0101800 [J.L.], 2017YFA0105001 [X.Z.], 2018YFE0204800 [J.L.]), Natural Science Foundation of Guangdong Province, China (2021A1515111156 [S.Y.]), Science and Technology Planning Project of Guangdong Province (2020B1212060052 [D.P.], 2019A1515011024 [S.C.]). Frontier Research Program of Bioland Laboratory (2018GZR110105012 [J.L.]). China Postdoctoral Science Foundation (2021000184 [C.Z.])

## Author contributions

S.Y. and J.L. designed the project; S.Y. and C.Z. performed the most experiments; J.H. performed the bioinformatics analysis with X.H., S.Ch., and G.Z; S.Y., C.Z., and S.W. isolated the primary EpiSC cell lines; S.Ca. and J.G. performed the blastocyst injection; H.L. and Y.Q. performed the scRNA-seq experiments; H.Z., X.W., B.C., L.X., and Y.W. performed the gene editing and RT-qPCR experiments; B.R. performed the ChIP-seq experiments; Z.Y. and Y.C. performed the PGCLC transplantation experiments; H.W. and Z.Z. provide suggestions to revise the manuscript; M.Z. and X.Z. supervised the germ cell experiments; F.L. and J.C. supervised the data analysis; J.L., D.P., and S.Ca. supervised the whole study, J.L., S.Y., and D.P. wrote the manuscript, J.L. approved the final version.

## Competing interests

The authors declare no competing interests.
