## [Peer Review File · Nature Communications]

BMP4 drives primed to naïve transition through PGC-like stateReviewers' comments:

Reviewer #1 (Remarks to the Author):

Yu et al. present detailed, single-cell analysis of a previously described primed to naïve reprogramming protocol and report that a) an early bifurcation is seen in cell fates, with one trajectory leading to correct naïve conversion, while the other results in cells which express some trophoblast markers, and b) the naïve branch transitions through a PGCLC intermediate, for which authors convincingly show functional PGCLC properties. This later finding is surprising and may help us better understand how germ cell fate specification is linked to the different flavors of pluripotent states.

There are however some important points that need to be clarified or revisited.

Major points:

-authors refer to placenta-like cells based on these cells expressing a handful of trophoblast-associated genes, such as *Krt8*, *Peg10*, *Plac1* and *Gata2*. Most of these genes are however expressed in other tissues as well. It would be more accurate to refer to these as "cells expressing some trophoblast markers".

A more comprehensive analysis of cell identity should also be performed by comparing gene expression profiles with multiple different embryonic and extraembryonic cell types – a reference dataset to compare to: Pijuan-Sala, B. et al. A single-cell molecular map of mouse gastrulation and early organogenesis. *Nature* 566, 490–495 (2019).

Finally, to test if these cells are indeed trophoblast-like, a functional assay should be performed by plating these cells (*c-kit*- or *Gata2*+) into trophoblast stem cell culture conditions. While the establishment of trophoblast stem cells is quite unlikely, the appearance of terminally differentiated trophoblast giant cells would support the trophoblast potential of this cell population.

-a previous report by Kime et al. (Kime, C. et al. Induced 2C expression and implantation-competent blastocyst-like cysts from primed pluripotent stem cells. *Stem Cell Rep*) demonstrates a somewhat different primed to naïve conversion protocol (with the commonality of using *Bmp4*) during which "induced blastocyst-like" structures emerge. These structures also contain cells that express certain trophoblast markers. Moreover Kime et al. demonstrated that *Prdm14* (a germ cell marker) is required for the formation of these blastocyst-like structures. This paper should be referenced and discussed.

-the expression profile of ESCs in Figure 1b, e and Figure 5d seems to be distinct from day8 reprogrammed cells. What accounts for this difference? Are these cells functionally naïve ESCs?

-While the data in Figure 3 d and e show the appearance of a PGCLC population during the course of differentiation, it does not show the dynamics of the transition of these cells into the naïve state. Could authors use a naïve-specific cell surface marker in their sorts to visualize this transition? If this is included, Figure 3 c and d could be moved to the Supplement.

-the cytokine dependency of emerging *Blimp* and *Stella* positive cells is interesting, however some information in Figure 3f could be made more clear – when during the differentiation process were cells moved into *Jaki* (how long were they cultured in it)? Is it correct to compare this plot to the day 6 plot in Figure 3e? Is the idea here that *SC+BV+* cells fail to exit the PGCLC state towards the naïve state, or that more *SC+BV+* cells are produced due to *STAT* inhibition?

-please provide more details on the experiment using *GK15* media and cytokines. Were day 3 cells plated into *GK15*? What is the purpose of trying with and without cytokines? In previous studies, which type of cells were used to specify PGCs in *GK15*? Is the observed efficiencies in this study (*GK15*~4.32%, *GK15*+cytokines~9.52%) comparable with previous studies?

-*Gata2* KO cells should be sorted for *c-kit* to demonstrate an early alteration of lineage trajectories.

-A key speculation that authors make in the discussion is that epigenetic inhibitors used in the reprogramming protocol may be responsible for allowing a PGCLC fate to emerge from primed pluripotent cells. This is an important point and could be easily tested by attempting reprogramming in the absence of individual or multiple epigenetic inhibitors. Would only the branch emerge that expresses some trophoblast markers?

Minor points:

-based on Figure 5d it would seem that separation of the 2 branches occurred already on day 1 (two population of blue cells in wt).

-How were day 6 BV+SC+ cells used in Figure 3g and 4 made? (as in Figure 3e, f or with GK15 media?)

-Figure 2 j could be grouped with Figure 3

-In Figure 3b a plot for Oct4 would be helpful to include

-"Day4/Day6 in vivo PGCLC" - should be in vitro?

-nuclear staining in Figure 4b would be helpful

-please add reference: "The primed EpiSCs, which corresponding to E6.5 epiblast, have lost the competence of PGC responsiveness (Ref)."

-the English of the manuscript needs improvement

--

Reviewer #2 (Remarks to the Author):

Yu et al. profile their BMP4 induced mouse primed to naive transition (Bi-PNT) system using scRNA-seq, and show by analyzing the resulting continuum of expression states there is an intermediate Prdm1+ PGC-like state. They then identify several genes via pseudotime analysis, some of which they validate via experimental analysis. Since I am not a biology domain expert, I cannot comment on the novelty of this study or its conceptual advance to the field. The scRNA-seq analysis is reasonably routine using standard software packages (harmony, seurat, scanpy, LineagePulse).

My main issue is that the way the paper is written, it is not clear for a general audience to glean why these findings are significant. For instance, the main finding of this paper appears to be a demonstration that cells transitioning through the Bi-PNT system appear to pass through an intermediate state that the authors call PGCLCs. What is the alternative hypothesis to make this a big deal? In other words, when cells transition from one expression state to another, it's not inconceivable that they will pass through intermediates. Is it significant because these cells are capable of spermatogenesis? I am wondering why the lack of spermatogenesis has not been shown in the knockout.

The one minor computational issue I have is that Fig. 1b does not indicate any biological replicates within time points. In the absence of that, it's not straightforward to know if there is a confound of batch effects.

I leave it to domain experts to judge the novelty or significance of these findings, but I would request that if this paper is accepted for publication, it be written in a way that draws in scientists who are not familiar with the field. As far as the computational analysis is concerned, it is sound but routine. That is to say that if the biological findings in this paper are considered significant, I have no issues with the computational analysis presented in this paper if it is considered worthy of publication by others.

--

Reviewer #3 (Remarks to the Author):

In a previous paper, the authors developed a method (BiPNT) for reverting EpiSCs to a more developmentally primitive naïve ESC-like state. In this paper, they use single cell RNA-sequencing to chart the path of reverting EpiSC to naïve lineage. The authors observe that some reverting cells express placental genes while others revert via a primordial germ cell-like intermediate. The authors further establish this point by showing that knockout of PRDM1, which is essential for germ lineage but dispensable for pluripotency, prevents reprogramming to a pluripotent state. By and large, the experiments are competently conducted and the paper is well written.

My biggest concern with the paper is whether this work is biologically significant. As the authors themselves note, is already known that treatment of EpiSCs with BMP4 can yield expression of placental genes (Bernardo 2011 Cell Stem Cell). It is also well established that early PGCs can give rise to cells very similar to embryonic stem cells (Resnick et al. "Long-term proliferation of mouse primordial germ cells in culture" 1992, Nature). And importantly, this is not a natural developmental process being studied. It's not even development in reverse. The authors establish that that a method they developed, BiPNT, produces a primed -> naïve reversion via a PGC-like intermediate, interesting but not something that happens in nature. There are some interesting biological ramifications to the fact that this particular set of inhibitors restores PGCLC competence for EpiSCs, implying a role for certain chromatin marks in "sealing off" PGC fate, but this point is not substantially explored. As such, it is probably better suited for a more specialized journal.

Other comments:

- In Figure 3, a lot of "PGC markers" are also expressed in naïve mESCs. These include Prdm1, Prdm14, Dppa3/Stella. Subsequent experiments in Figure 4 prove pretty convincingly that the PGCLCs identified by the authors are indeed PGCLCs, but the authors should be careful to establish that any "PGC marker" they identify is not expressed in ESCs, or else that they explain it is a dual marker.
- Figures 5d and 5e are hard to follow. It would be easier to compare the outcomes of the control and Prdm1 KO cells by just showing expression levels of key PGC, pluripotency and trophoblast genes like in 1c or 6e, rather than trying to match dots in different UMAPs.

Point-to-Point Rebuttal

Reviewers' comments:

Reviewer #1 (Remarks to the Author):

Yu et al. present detailed, single-cell analysis of a previously described primed to naïve reprogramming protocol and report that a) an early bifurcation is seen in cell fates, with one trajectory leading to correct naïve conversion, while the other results in cells which express some trophoblast markers, and b) the naïve branch transitions through a PGCLC intermediate, for which authors convincingly show functional PGCLC properties. This later finding is surprising and may help us better understand how germ cell fate specification is linked to the different flavors of pluripotent states.

There are however some important points that need to be clarified or revisited.

Major points:

-authors refer to placenta-like cells based on these cells expressing a handful of trophoblast-associated genes, such as Krt8, Peg10, Plac1 and Gata2. Most of these genes are however expressed in other tissues as well. It would be more accurate to refer to these as “cells expressing some trophoblast markers”.

Response: Thank this reviewer for her/his valuable suggestion. We have adjusted our description as suggested in the revised manuscript and termed these cells as trophoblast-like cells.

A more comprehensive analysis of cell identity should also be performed by comparing gene expression profiles with multiple different embryonic and extraembryonic cell types – a reference dataset to compare to: Pijuan-Sala, B. et al. A single-cell molecular map of mouse gastrulation and early organogenesis. Nature 566, 490–495 (2019).

Response: Thanks for this valuable suggestion. Accordingly, we downloaded the gene expression profiles as suggested¹, and further analyzed the cell types among the 954 trophoblast-like cells (D5 and D8), and showed with Sankey plot that 87.53% cells can be classified as ExE. Ectoderm, 7.97% cells are difficult to classify (5.14%, Rejected; 2.83%, Ambiguous), and very few cells can be classified as ExE. endoderm, ExE. mesoderm, or other cell types (**Supplementary Fig. 1b**).

Finally, to test if these cells are indeed trophoblast-like, a functional assay should be performed by plating these cells (c-kit- or Gata2+) into trophoblast stem cell culture conditions. While the establishment of trophoblast stem cells is quite unlikely, the appearance of terminally differentiated trophoblast giant cells would support the trophoblast potential of this cell population.

Response: We appreciate the reviewer for these valuable suggestions. With pleasure, we performed the experiments as suggested. Accordingly, we sorted the early *c-Kit* and *c-Kit*⁺ cells at Day 2 BiPNT, and further cultured them in trophoblast stem cells (TSC) maintain or differentiation medium. No TSC could be derived when cultured them in trophoblast stem cell (TSC) maintain medium (data not shown), just as mentioned by this reviewer. However, multi-nucleated giant cell can be observed in *c-Kit*_Diff_day6 group (Supplementary Fig. 2c) other than in *c-Kit*⁺_Diff_day6 group when those cells were cultured in TSC differentiation medium for 6 days (Supplementary Fig. 2b). Consistently, markers such as *PL1*, *PL2* (trophoblast giant cells markers), *Tpbpa* (spongiotrophoblast marker), and *Syna* (syncytiotrophoblast marker) were highly expressed in *c-Kit*_Diff_day6 cells specifically (Supplementary Fig. 2d). These data indicated that the early *c-Kit* cells are trophoblast-like.

-a previous report by Kime et al. (Kime, C. et al. Induced 2C expression and implantation-competent blastocyst-like cysts from primed pluripotent stem cells. Stem Cell Rep) demonstrates a somewhat different primed to naïve conversion protocol (with the commonality of using Bmp4) during which “induced blastocyst-like” structures emerge. These structures also contain cells that express certain trophoblast markers. Moreover Kime et al. demonstrated that Prdm14 (a germ cell marker) is required for the formation of these blastocyst-like structures. This paper should be referenced and discussed.

Response: Thanks for this constructive suggestion. In response, we have cited this paper² and discussed those points in revised manuscript (See discussion section).

-the expression profile of ESCs in Figure 1b, e and Figure 5d seems to be distinct from day8 reprogrammed cells. What accounts for this difference? Are these cells functionally naïve ESCs?

Response: We appreciate this reviewer for these kinds of questions. To answer these questions, we compared the gene expression profiles between Day 8 (D8) cells or naïve ESCs to Day 0 (D0) cells respectively (Supplementary Fig. 1c). And showed by Venn plot that 238/461 D8 specifically up/down regulated genes, and 370/319 naïve ESC specifically up/down regulated genes were divided (Supplementary Fig. 1d), which may lead to the distinction of naïve ESCs and D8 cells in original Figure 1b, e and Figure 5d. We further showed the GO terms of those genes (Supplementary Fig. 1e). In addition, we have shown previously that the colonies picked from Day8 are chimera and germline competent³, suggesting some D8 reprogrammed cells are functional equally to the naïve ESCs.

-While the data in Figure 3d and e show the appearance of a PGCLC population during the course of differentiation, it does not show the dynamics of the transition of these cells into the naïve state. Could authors use a naïve-specific cell surface marker in their

sorts to visualize this transition? If this is included, Figure 3 c and d could be moved to the Supplement.

Response: Thanks for the constructive suggestion by this reviewer. Accordingly, we compared the gene expression profiles among EpiSC, PGC and ESCs, and found a novel cell surface marker, CD105, which expressed specifically in naïve ESCs (Supplementary Fig. 3g). Using this surface marker, we showed by FACS analysis that a quite clear cell fate transition dynamics for the sorted Day6 BV⁺SC⁺ cells from PGCLCs to naïve state (Fig. 3f). We hope our new findings could satisfy the reviewer.

-the cytokine dependency of emerging Blimp and Stella positive cells is interesting, however some information in Figure 3f could be made more clearer – when during the differentiation process were cells moved into Jaki (how long were they cultured in it)?

Response: Thanks for the reviewer's suggestion. For detailed information about Figure 3f (New version Fig. 3e), we changed the medium to 2i+Jaki at Day3 during PNT, and followed with a further 3 days' culture (Day6-PNT) before FACS analysis. We also expanded relative information in the text of the revised manuscript.

Is it correct to compare this plot to the day 6 plot in Figure 3e?

Response: Yes. All the experiment system and analytical parameters of FACS are consistent between the original Fig. 3e (New version Fig. 3c) and Fig. 3f (New version Fig. 3e).

Is the idea here that SC⁺BV⁺ cells fail to exit the PGCLC state towards the naïve state, or that more SC⁺BV⁺ cells are produced due to STAT inhibition?

Response: We really appreciate this question. In previous report, the activation of LIF/STAT/JAK signaling pathway promotes the transition from PGC to naïve state⁴. So in here, we tempt to show that STAT inhibition block the exit from PGCLC towards naïve state during PNT, resulting in more BV⁺SC⁺ PGCLCs.

-please provide more details on the experiment using GK15 media and cytokines. Were day 3 cells plated into GK15? What is the purpose of trying with and without cytokines? In previous studies, which type of cells were used to specify PGCs in GK15? Is the observed efficiencies in this study (GK15~4.32%, GK15+cytokines~9.52%) comparable with previous studies?

Response: We appreciate these kind questions of the reviewer. In the revised manuscript, we made a simple graph to illustrate the performance of the experiment (Supplementary Fig. 3f). Indeed, GK15 medium (GK15; GMEM with 15% KSR) and cytokines were used for induction of PGCLCs from formative EpiLCs (Epibalst-like Cells)⁵. Here, we want to figure out whether those basic medium or cytokines can play any

role in PGCLCs induction during the second stage of BiPNT. And we showed that the efficiency of Day 6 PGCLCs induction with GK15+ cytokines in our system (GK15+ cytokines ~9.52%) is comparable with Hayashi's study (Day4-PGCLCs~13.5% and Day6-PGCLCs~7.2%).

-Gata2 KO cells should be sorted for c-kit to demonstrate an early alteration of lineage trajectories.

Response: We appreciate this suggestion by the reviewer. Accordingly, we analyzed *c-Kit*⁺ cells by FACS for Day3-PNT cells derived from two *Gata2*-KO cell lines, and showed that, *Gata2*-KO leads a significant increase of *c-Kit*⁺ population (WT~53%, KO1#~76.2%, KO2#~75.4%) (Supplementary Fig. 7d), RT-qPCR analysis further showed the maintain of PGC/Naïve markers in *Gata2*-KO *c-Kit*⁺ cells and the loss of trophoblast markers in *Gata2*-KO *c-Kit* cells (Supplementary Fig. 7e). These data indicated an early alteration of lineage trajectories after knockout of *Gata2*.

-A key speculation that authors make in the discussion is that epigenetic inhibitors used in the reprogramming protocol may be responsible for allowing a PGCLC fate to emerge from primed pluripotent cells. This is an important point and could be easily tested by attempting reprogramming in the absence of individual or multiple epigenetic inhibitors. Would only the branch emerge that expresses some trophoblast markers?

Response: Thanks for the constructive suggestion by this reviewer. Accordingly, we tested the reprogramming efficiency (PGCLC, BV⁺SC⁺%) by dropout individual inhibitor as suggested and showed that DOT1L inhibitor play crucial role for allowing PGCLC fate to emerge from primed EpiSCs (Fig. 6a, b; Supplementary Fig. 6a). The absence of DOT1L inhibitor blocks the expression of PGC makers almost totally as early as Day3 (Fig. 6b), and promotes the expression of trophoblast markers and increases *Gata2*-tdTomato⁺ cells at Day6 of BiPNT (Supplementary Fig. 6a, b). These data indicated that, DOT1L inhibition permits the emergence of PGCLCs branch, while its absence benefits the branch expressing some trophoblast markers at late BiPNT. Mechanically, as we showed in the revised manuscript, DOT1L inhibitor facilitates the loss of H3K79me2 from lineage factors *Gata3* and *Gata6*, and activates the PGC regulators such as *Nanog* and *Tfap2c*, which allowing the induction of PGCLCs from primed pluripotency (Fig. 6c-h, Supplementary Fig. 6a-e).

Minor points:

-based on Figure 5d it would seem that separation of the 2 branches occurred already on day 1 (two population of blue cells in wt).

Response: Yes, the cell population can be separated into 2 branches on day1 if analyzed carefully, and we adjusted our statement in the revised manuscript. Indeed, we use Harmony to generate two branches (trajectories) for the whole single cell dataset during BiPNT (Fig. 2a, b), and use UMAP to generate subpopulation for cells in Fig. 5d. UMAP has a high resolution for cell heterogeneity analysis and can

divide the day 1 cells into two subgroups.

-How were day 6 BV+SC+ cells used in Figure 3g and 4 made? (as in Figure 3e, f or with GK15 media?)

Response: The cells in Figure. 3g (New version Fig. 3d) and 4 are made the same as Figure. 3e (New version Fig. 3c).

-Figure 2 j could be grouped with Figure 3

Response: Thanks for this reviewer's suggestion, in the revised manuscript, we move the original Fig. 2j to Supplementary figure. 3a.

-In Figure 3b a plot for Oct4 would be helpful to include

Response: Thanks for this reviewer's suggestion, in the revised manuscript, the plot for Oct4 was included.

-“Day4/Day6 in vivo PGCLC” - should be in vitro?

Response: Yes, revised as suggested.

-nuclear staining in Figure 4b would be helpful

Response: Thanks for this reviewer's suggestion, in the revised version, the image of nuclear staining was included.

-please add reference: “The primed EpiSCs, which corresponding to E6.5 epiblast, have lost the competence of PGC responsiveness.”

Response: Thanks for this reviewer's suggestion, in the revised version, the references ^{5, 6} were cited in the right place.

-the English of the manuscript needs improvement

Response: We have sought professional editing. We hope the writing is acceptable for publication now.

--

Reviewer #2 (Remarks to the Author):

Yu et al. profile their BMP4 induced mouse primed to naive transition (Bi-PNT) system using scRNA-seq, and show by analyzing the resulting continuum of expression states there is an intermediate Prdm1+ PGC-like state. They then identify several genes via pseudotime analysis, some of which they validate via experimental analysis. Since I am not a biology domain expert, I cannot comment on the novelty of this study or its conceptual advance to the field. The scRNA-seq analysis is reasonably routine using standard software packages (harmony, seurat, scanpy, LineagePulse).

My main issue is that the way the paper is written, it is not clear for a general audience to glean why these findings are significant. For instance, the main finding of this paper appears to be a demonstration that cells transitioning through the Bi-PNT system appear

to pass through an intermediate state that the authors call PGCLCs. What is the alternative hypothesis to make this a big deal? In other words, when cells transition from one expression state to another, it's not inconceivable that they will pass through intermediates. Is it significant because these cells are capable of spermatogenesis? I am wondering why the lack of spermatogenesis has not been shown in the knockout.

Response: We appreciate the valuable suggestions from this reviewer. Indeed, the main purpose for this manuscript is to reveal the existence of a spermatogenesis competent intermediate PGCLC during the process of BMP4 induced Primed-Naïve Transition (BiPNT) we report recently ³ by single cell analysis. We fully agree with the reviewer that, the transition from one cell state to another will conceivably pass through intermediates. However, it is always unpredictable which kind of intermediate, especially the one has corresponding counterpart in vivo, could emergent during cell fate transition, which can inspire new insight to understand the relationship or connection among various cell fates during cell fate transitions in vitro. For example, previous study has revealed a XEN (extraembryonic endoderm)-like state mediates mouse chemical reprogramming ⁷ or a trophectoderm-like state exists during human somatic reprogramming ⁸, which offered new insights to understand the cell fate landscapes during somatic cell reprogramming. Here, very surprisingly, we found the existence of a PGC (the precursor of sperm or oocyte)-like intermediate governing the BiPNT process in vitro, for which, we think, implied several quite important significant in biology. Firstly, primed pluripotent stem cells (EpiSCs) have been proved difficultly to be induced into PGC fate by BMP4 stimulation ⁵, our work breaks through this technical bottleneck. Secondly, Primed-PGCLC transition (PP_GT) established a robust method for generating PGCLC from EpiSCs which can be used as an ideal cell model in vitro for the study of reproductive development and spermatogenesis.

The one minor computational issue I have is that Fig. 1b does not indicate any biological replicates within time points. In the absence of that, it's not straightforward to know if there is a confound of batch effects.

Response: We thanks the reviewer for his/her kindly suggestion. Regarding to the batch effects, BiPNT is a very robust and efficient (~80%) reprogramming system with less varies in batches. Rationally, we collected sufficient cells for single-cell analysis at each time point (D0, D1, D2, D3, D5, D8) for the capturing of the dynamics and subpopulation during BiPNT. We are quite confident that batch effect, if it exists, would not affect the main conclusions or findings of our work.

I leave it to domain experts to judge the novelty or significance of these findings, but I would request that if this paper is accepted for publication, it be written in a way that draws in scientists who are not familiar with the field. As far as the computational analysis is concerned, it is sound but routine. That is to say that if the biological findings in this paper are considered significant, I have no issues with the computational analysis presented in this paper if it is considered worthy of publication by others.

Response: We are very grateful for this reviewer's suggestions. Accordingly, we have made some adjustment on the writing style, and also sought professional editing to improve the language of our manuscript. We hope these efforts can make our work more accessible to broad readers including the ones not familiar with the field.

--

Reviewer #3 (Remarks to the Author):

In a previous paper, the authors developed a method (BiPNT) for reverting EpiSCs to a more developmentally primitive naïve ESC-like state. In this paper, they use single cell RNA-sequencing to chart the path of reverting EpiSC to naïve lineage. The authors observe that some reverting cells express placental genes while others revert via a primordial germ cell-like intermediate. The authors further establish this point by showing that knockout of PRDM1, which is essential for germ lineage but dispensable for pluripotency, prevents reprogramming to a pluripotent state. By and large, the experiments are competently conducted and the paper is well written.

My biggest concern with the paper is whether this work is biologically significant. As the authors themselves note, is already known that treatment of EpiSCs with BMP4 can yield expression of placental genes (Bernardo 2011 Cell Stem Cell). It is also well established that early PGCs can give rise to cells very similar to embryonic stem cells (Resnick et al. "Long-term proliferation of mouse primordial germ cells in culture" 1992, Nature). And importantly, this is not a natural developmental process being studied. It's not even development in reverse. The authors establish that that a method they developed, BiPNT, produces a primed -> naïve reversion via a PGC-like intermediate, interesting but not something that happens in nature. There are some interesting biological ramifications to the fact that this particular set of inhibitors restores PGCLC competence for EpiSCs, implying a role for certain chromatin marks in "sealing off" PGC fate, but this point is not substantially explored. As such, it is probably better suited for a more specialized journal.

Response: We thank this reviewer for his/her positive comments and valuable suggestions. To the concern of the biological significance for our work, firstly, we'd like to claim that we established a robust method to restores PGCLC competence for EpiSCs, which could not be achieved in previous study⁵, and can be used as a cellular model for in vitro reproductive development in future. Secondly, according to the suggestion by this reviewer, in the revised manuscript, we further investigated the underlying mechanism within restoring PGCLC competence for EpiSCs by inhibitors. By using a series of biochemical and multiple omics analysis, including ChIP-seq, RNA-seq and ATAC-seq, very luckily, as pointed by this reviewer, we found DOT1L mediated H3K79me2 plays a key role in "sealing off" PGC fate from EpiSCs (Fig. 6a-i). In detail, DOT1L inhibition decreases the global H3K79me2, represses the expression of lineage factors *Gata3* and *Gata6*, while activates the PGC regulators *Nanog* and *Tfap2c*,

resulting in the successful induction of PGCLCs from primed EpiSCs. Finally, although Primed-> PGCLC->Naïve transition is not a nature developmental process, as mentioned, deep study on this process could reveal key events or mechanisms mirror the early embryonic development, for example, the function of H3K79 methylation in lineage specification, especially the PGC specification. Therefore, we sincerely hope that those biological significances represented in the revised manuscript can support our work suitable for publication in Nature Communications.

Other comments:

- In Figure 3, a lot of “PGC markers” are also expressed in naïve mESCs. These include *Prdm1*, *Prdm14*, *Dppa3/Stella*. Subsequent experiments in Figure 4 prove pretty convincingly that the PGCLCs identified by the authors are indeed PGCLCs, but the authors should be careful to establish that any “PGC marker” they identify is not expressed in ESCs, or else that they explain it is a dual marker.

Response: Thanks for the reviewer's suggestion. Indeed, PGC and ESC share many common molecular markers, in the revised manuscript, we strictly distinguish the only PGC specific markers (such as *Prdm1*) or only ESC specific markers (such as *Klf4*), and other genes such as *Prdm14*, *Dppa3/Stella* were classified as PGC/Naïve pluripotency markers.

- Figures 5d and 5e are hard to follow. It would be easier to compare the outcomes of the control and *Prdm1* KO cells by just showing expression levels of key PGC, pluripotency and trophoblast genes like in 1c or 6e, rather than trying to match dots in different UMAPs.

Response: We appreciate the kind suggestion from this reviewer. In response to the reviewer's suggestion, original Figure 5e was replaced with new version Fig. 5e, f.

References

1. Pijuan-Sala, B. *et al.* A single-cell molecular map of mouse gastrulation and early organogenesis. *Nature* **566**, 490-495 (2019).
2. Kime, C. *et al.* Induced 2C Expression and Implantation-Competent Blastocyst-like Cysts from Primed Pluripotent Stem Cells. *Stem Cell Reports* **13**, 485-498 (2019).
3. Yu, S. *et al.* BMP4 resets mouse epiblast stem cells to naïve pluripotency through ZBTB7A/B-mediated chromatin remodelling. *Nat Cell Biol* **22**, 651-662 (2020).
4. Leitch, H.G. *et al.* Rebuilding pluripotency from primordial germ cells. *Stem Cell Reports* **1**, 66-78 (2013).
5. Hayashi, K., Ohta, H., Kurimoto, K., Aramaki, S. & Saitou, M. Reconstitution of the mouse germ cell specification pathway in culture by pluripotent stem cells. *Cell* **146**, 519-532 (2011).
6. Ohinata, Y. *et al.* A signaling principle for the specification of the germ cell lineage in mice. *Cell* **137**, 571-584 (2009).
7. Zhao, Y. *et al.* A XEN-like State Bridges Somatic Cells to Pluripotency during Chemical

Reprogramming. *Cell* **163**, 1678-1691 (2015).

8. Liu, X. *et al.* Reprogramming roadmap reveals route to human induced trophoblast stem cells. *Nature* **586**, 101-107 (2020).

REVIEWER COMMENTS

Reviewer #3 (Remarks to the Author):

In my initial review, I did not feel the paper warranted publication in Nature Communications because there was insufficient biology. With the new experiments demonstrating that DOT1L is important for shutting off germ cell lineage in EpiSCs, I change my view and regard the paper as sufficiently strong. The revisions have also made it more comprehensible, and addressed the other concerns I had. The most significant limitation is that the data linking H3K79me2 to placental gene expression is fairly poor. Yes Gata3 and Gata6 have H3K79me2 at their promoters and gain H3K27me3 when DOT1L is inhibited, but presumably thousands of genes lose H3K79me2. Is there evidence of a systematic drop in gene expression in genes with lots of H3K79me2 (or alternatively, that genes that lose expression upon DOT1L inhibition have elevated H3K79me2?). Do the germline genes have less starting H3K79me2? I don't think it's absolutely essential to establish such correspondences, but the authors should check and report a result either way. They should also avoid being too confident that Gata3 and Gata6 are direct DOT1L regulatory targets.

Other comments:

- Given the observation that amnion is similar to and frequently mistaken for trophoblast (see for example Guo et al. "Human naïve epiblast cells possess unrestricted lineage potential" Cell Stem Cell) it would be beneficial to explicitly show lack of amnion markers in their data as a supplementary figure. I suspect this is the case, since the Pijuan-Sala et al. data includes amnion and the authors see a closer resemblance to trophoblast, but it would be good to show explicitly.
- In Figure 6c, the upregulation of Hox genes in DOT1Li conditions is quite interesting. Are these expressed in trophoblast? If not, is it possible that a different cell type is forming (presumably a somatic lineage of some sort)?
- Line 389: "These data indicated that DOT1Li promotes PPGT through the activation of Nanog or Tfap2c, at least partially." This is not supported by Supplementary Figure 6d, which shows that DOT1L improves germ cell formation even with Nanog overexpression (if DOT1L acted mainly through Nanog upregulation, you would not see an additive effect). Probably this sentence should be eliminated.
- This manuscript has some grammar and language issues, although none very serious.

Reviewer #4 (Remarks to the Author):

In the present manuscript, Yu and colleagues investigate the in vitro transition from a primed to naïve pluripotent state building upon on their previous published BiPNT protocol (Yu et al. Nat Cell Bio 2020). This consists of culturing EpiSCs in 2 different media - BMP4+DOLT1i+EZH2i for 3 days, followed by 2iL medium thereafter. The authors present scRNA-seq analysis during this primed to naïve transition, finding two different fate choices; one towards the naïve state and the other one towards "trophoblast-like" state. The main focus of the paper is on the naïve pathway. Further gene expression analysis shows upregulation of PGC-related genes during the conversion process. However, interpretation of this data is complicated by the significant overlap between 'PGC genes' and 'naïve genes'. The authors present two key experiments that support their view that BiPNT does involved transition through a PGC-like intermediate. Firstly, these PGC-like intermediate cells appear to be able to repopulate testes that are devoid of germ cells. Secondly, Prdm1-KO ES cells do not appear to be able to undergo BiPNT. By way of mechanism, the authors mainly investigate the role of DOTL1 inhibition and present a preliminary model that is consistent with their observations.

This manuscript has at its core data that is both surprising and fascinating, and would certainly be of interest to the germ cell, pluripotent stem cell, epigenetics and reprogramming fields. The authors have done a lot of work, including many well-conducted experiments. However, the manuscript still needs significant work before it is ready for publication in Nature Communications.

(Please note: I have been added for the second revision, having not taken part in the first round of reviews).

Major issues.

1. Terminology and clarity

The authors are hampered by their inconsistent use of terminology around pluripotent states, and also the often unclear descriptions of the experimental setups and systems. For instance,

a) Naïve vs PGC states. A longstanding problem for the field is that almost all 'naïve genes' are expressed in PGCs (with the exception of *Klf4* and *Tbx3*) and that the transcripts of almost all PGC genes can be found in naïve ES cells (for instance, *Prdm1* transcript can be detected in many studies, although *Prdm1* protein is not readily detected in naïve cultures). However, for the most part in the figures and text the authors simply ignore this issue and make arbitrary decisions as to which group to assign a certain gene. For instance, Figure 3a (and 5f) – none of the genes in the 'PGC (early)' cluster are specific to PGCs and so this designation seems completely arbitrary. Why is *Nanog* not considered a PGC (early) gene? It has a stronger PGC phenotype than *Dppa3* and *Ifitm1-3*. Also *c-Kit* (classically thought of as a PGC-marker) is used as a 'naïve branch' marker. In reality, what the authors appear to be observing is the upregulation of the naïve pluripotency network as part of PGCLC generation – as their *Prdm1* KO experiments appear to suggest this is essential (see below). Why not just state this from the start, which may have added benefit of allowing them to significantly shorten the manuscript.

b) Primed state. In Figure 3b *Oct4* is designated a primed marker – this is clearly incorrect. Again, I would rather avoid these arbitrary designations and concentrate on describing what the data actually shows – this may allow for significant cuts and a shorter manuscript.

c) *Oct4*-GFP. This reporter and its properties are not well described, and there is no clear reference to which *Oct4*-GFP reporter they are using. They describe this as a 'naïve signal' (Line 137). However, a knock-in reporter or transgenic reporter with the full *Oct4* promoter would be expressed in both naïve and primed cells (and PGCs). This is particularly important for interpretation in Lines 229-232. Is there an emergence of *Oct4*-GFP+/KLF4- cells, or is the starting point (EpiSCs), not *Oct*-GFP+/KLF4-? Currently the authors do not clearly present data that distinguishes the identity of these cells (again the terminology and clarity of explanation does not help). Furthermore, the data for *Oct4*-GFP and *Klf4* requires quantification (as Supplementary Figure 3b clearly shows the presence of double and single positive cells at both time points).

d) Double reporter cell line: Figure 1G is insufficient to conclude in Line 138 that the two cells types are 'incompatible and mutually exclusive'. Instead, a well-controlled FACS analysis (with experimental replicates) is needed to backup such a strong statement.

2. *Prdm1* KO experiments.

This experiment seems to provide compelling data that, unlike other conversion paradigms, BiPNT requires transition through a PGC-like intermediate state. However, the results of these experiments are not completely clear. The authors state that 'no GFP+ cells can be generated from *Prdm1*^{-/-} cells'. Does this mean that no colonies are obtained and that it is not possible to generate any naïve pluripotent stem cells during BiPNT? If so, this should be made clear as it is a very important point. There was previous significant controversy around the idea that reversion of EpiSCs to so-called 'rESCs' (PMID: 22770244) might involve a transition through a PGC-like state. The authors of the previous study used the same assay (*Prdm1* KO cells) to show unequivocally that this conversion can occur independent of a germ cell intermediate. The opposite finding for BiPNT, that conversion is completely dependent on this transition, is of great interest. However, it does mean that the title needs to be changed to something less general – as not all mouse primed to naïve transitions require PGC-like intermediates.

I also have some further technical concerns. What does Supplementary Figure 5b show? PRDM1 protein would not normally be detected in ES cells, and the cell type is not described. Western blot analysis in a PRDM1 expressing cell type derived from the KO cells should be shown to confirm lack of PRDM1 protein in knockout cells (ideally WT and rescue cells would be used as controls). Sequencing data to show the disrupted locus would also normally be shown. How was the rescued performed? The only information I could find was that this was *Prdm1* overexpression. This would not normally be

tolerated by naïve cells (see PMID: 23851488 in which it is clearly stated that ES cells do not tolerate PRDM1 expression). I therefore find it difficult to understand what the GFP positive cells in Figure 5b and Supplementary Figure 5d are? Was this a conditional overexpression system to allow survival of these cells? Where stable Prdm1 overexpressing cell lines obtained?

3. PGCLC experiments. The emergence of PGCLCs during BiPNT is overall quite well described and demonstrated. Most importantly the functional experiments show testes repopulation. As mentioned above Figure 3b should be presented with different headings that don't include arbitrary gene designations. Figure 3c and 3d seem very clear. However, subsequent experiments become somewhat muddled.

a) Figure 3f. Why are PGCLCs induced in GK15 media alone? This is a very surprising result. In contrast, the GK15+cytokines experiment does seem an important finding – this is the clearest demonstration that initiating BiPNT re-establishes germline competence of EpiSC. Perhaps a side-by-side experiment with untreated EpiSCs would be even more clear-cut?

b) The CD105 experiments (for instance 3f) are difficult to interpret. The FACS plot seems to indicate that at least some EpiSCs will spontaneously upregulate this marker when plated in 2iL? Why not just perform an immunostaining for Klf4? This should very clearly show transition from a PGCLC state to an ESC state – and is a validated marker.

c) H19 – although there is some loss of methylation, it is not at all clear that imprint erasure has occurred (and indeed this would not necessarily be expected in PGCLCs). Perhaps the authors are measuring the effect of 2iL to reduce DNA methylation more globally? As Day 6 BVSC cells have presumably been exposed to 2iL for 3 days?

d) Chimera experiments. More careful explanation of the findings is needed here. Although it is not consistently stated, it appears the injected BV+SC+ cells are from Day 6 cultures (meaning they have already been exposed to 2iL for 3 days?). How exactly were in BV+SC+ cells passaged in 2iL treated? On what day were they passaged and replated, and how much longer were they cultured for? How do the authors interpret the formation of a single chimaera for the BV+SC+ cells? Is this due to contamination with naïve cells? Or due to rapid conversion of some PGCLCs to naïve pluripotency in their protocol (which does expose PGCLCs to 2iL).

Finally taking my points 2 and 3 together. If the BiPNT transition requires an obligate route through a PGCLC state – then does gating out all the BV+SC+ positive cells on Day 6 (and/or Day 8) completely prevent the formation of pluripotent stem cells clones? For instance, can the BVSC single or double negative cells in Figure 3f form naïve pluripotent stem cells (without themselves going through a BV+SC +positive state)?

3. Testes repopulation. This is important data and looks quite clear cut. Ideally an uninjected control should be shown in Supplementary Figure 4. It seems quite clear that repopulation occurs, and I don't necessarily think further experiments are required. However, the authors should comment on whether they attempted to test the functionality of any sperm/spermatids. Were any teratoma observed? (Perhaps indicating that early transition to naïve pluripotency does occur in the BiPNT protocol).

4. Mechanistic aspects. The model presented for the impact of DOT1L inhibition is clearly incomplete, but there are some interesting experiments and observations. This is already an extensive manuscript, and so I do not think that a complete mechanistic dissection is necessarily required. However, I think the authors should focus on more clearly explaining the experiments they have done, and emphasising what is and is not known, rather than trying to present the model as complete. For instance, it is not clear why DOTL inhibition leads to upregulation of Nanog or Tfap2c and whether this is related to changes in H3K79methylation at these loci? A direct connection between alterations in expression of Gata3/6 and of Nanog (and other PGC genes) in generating PGCLCs in BiPNT is really not clear. This is perhaps also due to the experiments being presented in a slightly idiosyncratic order. Also, it is not clear why the focus changes from Gata3/6 to Gata2. If the main function of DOT1L inhibition is to reduce activation of Gata factors during BiPNT, then presumably knockout of these factors should reduce the requirement for DOT1L inhibition?

5. EG cell-like cells (EGCLCs). Should the pluripotent stem cell lines derived using the BiPNT protocol actually be referred to as EGCLCs, given they apparently all go through a PGCLC state? Does Jak inhibition block the formation of naïve pluripotent stem cells/EGCLCs, as would be anticipated from ref 31 (PMID: 24052943)

Minor points

1. Introduction. Line 62. This is not an adequate definition of pluripotency (the capacity to generate all the cells types in an individual must be the property of a single cell).
2. Introduction. Line 65-66. The division between ESC and EpiSC is confusing here. Are they really so similar by gene expression profile? Their in vitro differentiation potentials are also different – as typically EpiSCs do not efficiently give rise to PGCs. I would rework this section as it is a bit contradictory to the rest of the manuscript.
3. Line 70-71 – unclear English
4. Line 81-82 – unclear English
5. Line 133 – ‘Naïve ones’ – meaning unclear.
6. Line 145 – seems to include a change of font?
7. Line 162 – imprinted genes
8. Supplementary Figure 2c (and line 181-183). The big, flat cells shown are not obviously trophoblast giant cells (TGCs) and there is insufficient data to indicate successful generation of TGCs. I would just omit this – it really is not an important aspect of the paper. The exact trophoblast potential of these cells could surely form part of a future manuscript.
9. Supplementary Figure 3e is difficult to interpret as it is performed on bulk cultures. In addition, the impact of these factors on PGC conversion to pluripotency is highly context dependent as shown by ref 31 (PMID: 24052943) and PMID: 32944903.
10. Line 248-249 – incomplete sentence, please correct
11. Line 301 – chimeras
12. Line 311 – the authors have not shown the PGCLCs produce functional germ cells. This statement should be altered unless new data is available.
13. Line 458. Spelling of blastocyst
14. Zbtb7a/b. If this data is to be included, it should be in the Results section and properly incorporated into the narrative.

REVIEWER COMMENTS

Reviewer #3 (Remarks to the Author):

In my initial review, I did not feel the paper warranted publication in Nature Communications because there was insufficient biology. With the new experiments demonstrating that DOT1L is important for shutting off germ cell lineage in EpiSCs, I change my view and regard the paper as sufficiently strong. The revisions have also made it more comprehensible and addressed the other concerns I had.

The most significant limitation is that the data linking H3K79me2 to placental gene expression is fairly poor. Yes Gata3 and Gata6 have H3K79me2 at their promoters and gain H3K27me3 when DOT1L is inhibited, but presumably thousands of genes lose H3K79me2. Is there evidence of a systematic drop in gene expression in genes with lots of H3K79me2 (or alternatively, that genes that lose expression upon DOT1L inhibition have elevated H3K79me2?). Do the germline genes have less starting H3K79me2? I don't think it's absolutely essential to establish such correspondences, but the authors should check and report a result either way. They should also avoid being too confident that Gata3 and Gata6 are direct DOT1L regulatory targets.

Response: Thanks to the reviewer for these valuable suggestions. Generally, the methylation of H3K79 is associated with active chromatin and transcription elongation¹, thus was usually considered as an active signal for gene expression. In BiPNT, DOT1L inhibition almost abolishes H3K79me2 genomic widely as measured by reference-normalized ChIP-seq (Rebuttal Fig. 1a or Fig. 6g). When analyzing the correlation between H3K79me2 and gene expression in BiPNT, we found that 30.6% up-regulated genes, 58.0% down-regulated genes, and 79.1% unchanged genes are marked by H3K79me2, respectively (Rebuttal Fig. 1b, c or Fig. 6c, h). For genes such as *Gata3/6*, we observed a loss of H3K79me2 in the promoter region and gene body, and consequently a decrease in mRNA upon DOT1L inhibition at Day3 (Rebuttal Fig. 1d or Fig. 6i), consistent with the role for H3K79me2 in active transcription¹. For genes associated with germ cell, such as *Nanog*, *Prdm1*, *Tfap2c* and *Nanos3*, we observed a loss of H3K79me2 in their gene body region, while an increase in mRNA upon the inhibition of DOT1L (Rebuttal Fig. 1d or Fig. 6i). To determine whether *Gata3/6* deficiency could reduce the requirement of DOT1Li, we generated *Gata3* or *Gata6* knock-out EpiSCs (Rebuttal Fig. 1e-h or Supplementary Fig. 6g-j) and found that *Gata3/6* KO can promote the generation of BV⁺SC⁺ cells in the absence of DOT1Li (Rebuttal Fig. 1i, j or Fig. 6j, k). In addition, the presence of DOT1Li can further enhance the efficiency of BV⁺SC⁺ cell induction in *Gata3* or *Gata6* KO EpiSCs, comparing with WT EpiSCs (Rebuttal Fig. 1j or Fig. 6k). These data indicate that *Gata3* and *Gata6* are DOT1Li targets during BV⁺SC⁺ cell induction from EpiSCs.

Rebuttal Fig. 1

Other comments:

- Given the observation that amnion is similar to and frequently mistaken for trophoblast (see for example Guo et al. "Human naïve epiblast cells possess unrestricted lineage potential" Cell Stem Cell) it would be beneficial to explicitly show lack of amnion markers in their data as a supplementary figure. I suspect this is the case, since the Pijuan-Sala et al. data includes amnion and the authors see a closer resemblance to trophoblast, but it would be good to show explicitly.

Response: Thanks for these constructive suggestions. Accordingly, in the revised manuscript, we show the data including the expression pattern of amnion makers *Is1* and *Igfbp3* as well as trophoblast marker *Elf5* by heatmap (Rebuttal Fig. 2 or Supplementary

Fig. 1d). The result indicates the amnion markers were not expressed in *Elf5* positive trophoblast-like cells, which can distinguish them from the amnion cells.

Rebuttal Fig. 2

- In Figure 6c, the upregulation of Hox genes in DOT1Li conditions is quite interesting. Are these expressed in trophoblast? If not, is it possible that a different cell type is forming (presumably a somatic lineage of some sort)?

Response: Thanks for these comments. In Fig. 6c, when DOT1Li withdrawing, most of the Hox genes were upregulated at Day3. We re-analyzed the scRNA-seq data of BiPNT and found that most of the Hox family genes are not expressed in trophoblast branch, and seemly co-expressed with somatic markers, such as *T* and *Evx1* in the early stage of BiPNT (Rebuttal Fig. 3). Therefore, we could not exclude the forming of other cell type(s) such as somatic lineage in the absence of DOT1L inhibition.

Rebuttal Fig. 3

- Line 389: "These data indicated that DOT1Li promotes PPGT through the activation of *Nanog* or *Tfap2c*, at least partially." This is not supported by Supplementary Figure 6d, which shows that DOT1L improves germ cell formation even with *Nanog* overexpression (if DOT1L acted mainly through *Nanog* upregulation, you would not see an additive effect). Probably this sentence should be eliminated.

Response: We agree with this opinion and eliminate this sentence as suggested in the revised manuscript.

- *This manuscript has some grammar and language issues, although none very serious.*

Response: We thank the reviewer for this kind suggestion and have sought for professional help in grammar and language issues in the revised manuscript.

Reviewer #4 (Remarks to the Author):

In the present manuscript, Yu and colleagues investigate the in vitro transition from a primed to naïve pluripotent state building upon on their previous published BiPNT protocol (Yu et al. Nat Cell Bio 2020). This consists of culturing EpiSCs in 2 different media - BMP4+DOLT1i+EZH2i for 3 days, followed by 2iL medium thereafter. The authors present scRNA-seq analysis during this primed to naïve transition, finding two different fate choices; one towards the naïve state and the other one towards “trophoblast-like” state. The main focus of the paper is on the naïve pathway. Further gene expression analysis shows upregulation of PGC-related genes during the conversion process. However, interpretation of this data is complicated by the significant overlap between ‘PGC genes’ and ‘naïve genes’. The authors present two key experiments that support their view that BiPNT does involved transition through a PGC-like intermediate. Firstly, these PGC-like intermediate cells appear to be able to repopulate testes that are devoid of germ cells. Secondly, Prdm1-KO ES cells do not appear to be able to undergo BiPNT. By way of mechanism, the authors mainly investigate the role of DOTL1 inhibition and present a preliminary model that is consistent with their observations.

This manuscript has at its core data that is both surprising and fascinating, and would certainly be of interest to the germ cell, pluripotent stem cell, epigenetics and reprogramming fields. The authors have done a lot of work, including many well-conducted experiments. However, the manuscript still needs significant work before it is ready for publication in Nature Communications.

Response: We thank this reviewer for her/his appreciation of our works.

(Please note: I have been added for the second revision, having not taken part in the first round of reviews).

Major issues.

1. Terminology and clarity

The authors are hampered by their inconsistent use of terminology around pluripotent states, and also the often unclear descriptions of the experimental setups and systems. For instance,

a) Naïve vs PGC states. A longstanding problem for the field is that almost all 'naïve genes' are expressed in PGCs (with the exception of *Klf4* and *Tbx3*) and that the transcripts of almost all PGC genes can be found in naïve ES cells (for instance, *Prdm1* transcript can be detected in many studies, although *Prdm1* protein is not readily detected in naïve cultures). However, for the most part in the figures and text the authors simply ignore this issue and make arbitrary decisions as to which group to assign a certain gene. For instance, Figure 3a (and 5f)– none of the genes in the 'PGC (early)' cluster are specific to PGCs and so this designation seems completely arbitrary. Why is *Nanog* not considered a PGC (early) gene? It has a stronger PGC phenotype than *Dppa3* and *Ifitm1-3*. Also *c-Kit* (classically thought of as a PGC-marker) is used as a 'naïve branch' marker. In reality, what the authors appear to be observing is the upregulation of the naïve pluripotency network as part of PGCLC generation – as their *Prdm1* KO experiments appear to suggest this is essential (see below). Why not just state this from the start, which may have added benefit of allowing them to significantly shorten the manuscript.

Response: We thank this reviewer for these valuable suggestions. Accordingly, we defined the use of Naïve markers or PGC markers as suggested in the revised manuscript (Fig. 3a and 5e) to avoid confusion. Previously, we represented our story beginning with primed to naïve transition and then focused on PGC intermediate, the real logic of our work. We can change the logic of the manuscript if necessary. Indeed, we have tried to shorten the manuscript and welcome more suggestions from the reviewers and editor.

b) Primed state. In Figure 3b *Oct4* is designated a primed marker – this is clearly incorrect. Again, I would rather avoid these arbitrary designations and concentrate on describing what the data actually shows – this may allow for significant cuts and a shorter manuscript.

Response: We thank this reviewer for the valuable suggestion. Accordingly, we replaced *Oct4* with *Fgf5*, a well-known primed pluripotency gene, to avoid arbitrary designations in the revised manuscript (Fig. 3b).

c) *Oct4*-GFP. This reporter and its properties are not well described, and there is no clear reference to which *Oct4*-GFP reporter they are using. They describe this as a 'naïve signal' (Line 137). However, a knock-in reporter or transgenic reporter with the full *Oct4* promoter would be expressed in both naïve and primed cells (and PGCs). This is particularly important for interpretation in Lines 229-232. Is there an emergence of *Oct4*-GFP+/KLF4- cells, or is the starting point (EpiSCs), not *Oct*-GFP+/KLF4-? Currently the authors do not clearly present data that distinguishes the identity of these cells (again the terminology and clarity of explanation does not help). Furthermore, the data for *Oct4*-GFP and *Klf4* requires quantification (as Supplementary Figure 3b clearly shows the presence of double and single positive cells at both time points).

Response: We are sorry about the unclear statement in the former manuscript. Indeed, the transgenic *Oct4*-GFP reporter used here is driven by a cassette lacking the proximal enhancer (Δ PE) of *Oct4*². In this situation, GFP signal can be activated in naïve ESCs or PGCs, but not in primed EpiSCs^{2, 3}. We have cited the reference and changed all the “*Oct4*-GFP” into “ Δ PE-*Oct4*-GFP” in the revised manuscript. Therefore, the emergence of Δ PE-*Oct4*-GFP⁺/KLF4⁻ cells are not the starting EpiSCs, and may be PGCLCs. As suggested by the reviewer, we quantified the cells by Δ PE-*Oct4*-GFP and KLF4, and show that 16.07% Δ PE-*Oct4*-GFP⁺/KLF4⁻ and 6.54% Δ PE-*Oct4*-GFP⁺/KLF4⁺ cells were detected at day6, 7.64% Δ PE-*Oct4*-GFP⁺/KLF4⁻ and 71.46% Δ PE-*Oct4*-GFP⁺/KLF4⁺ cells, were detected at day8 (Rebuttal Fig. 4 or Supplementary Fig. 3b).

Rebuttal Fig. 4

d) Double reporter cell line: Figure 1G is insufficient to conclude in Line 138 that the two cells types are ‘incompatible and mutually exclusive’. Instead, a well-controlled FACS analysis (with experimental replicates) is needed to backup such a strong statement.

Response: We thank the reviewer for this valuable suggestion. Accordingly, we performed FACS experiment as suggested, and showed a mutually exclusive for *Gata2*-tdTomato signal and Δ PE-*Oct4*-GFP signal during BiPNT (Rebuttal Fig. 5 or Supplementary Fig. 1b).

Rebuttal Fig. 5

2. Prdm1 KO experiments. This experiment seems to provide compelling data that, unlike other conversion paradigms, BiPNT requires transition through a PGC-like intermediate state. However, the results of these experiments are not completely clear. The authors

state that 'no GFP+ cells can be generated from Prdm1^{-/-} cells'. Does this mean that no colonies are obtained and that it is not possible to generate any naïve pluripotent stem cells during BiPNT? If so, this should be made clear as it is a very important point. There was previous significant controversy around the idea that reversion of EpiSCs to so-called 'rESCs' (PMID: 22770244) might involve a transition through a PGC-like state. The authors of the previous study used the same assay (Prdm1 KO cells) to show unequivocally that this conversion can occur independent of a germ cell intermediate. The opposite finding for BiPNT, that conversion is completely dependent on this transition, is of great interest. However, it does mean that the title needs to be changed to something less general – as not all mouse primed to naïve transitions require PGC-like intermediates.

Response: Thanks to the reviewer's valuable proposals. "no GFP+ cells" mentioned here means that it's not possible to generate PGCLC or naïve stem cell, as both PGC or naïve cell are GFP^{+2, 3}. We agree with this reviewer's opinion that, not all mouse primed to naïve transitions undergo PGC-like intermediates, and we have changed our title to "Single Cell Sequencing Reveals Early PGC-like Intermediates During BMP4 Driven Mouse Primed to Naïve Transition" for consideration.

I also have some further technical concerns. What does Supplementary Figure 5b show? PRDM1 protein would not normally be detected in ES cells, and the cell type is not described. Western blot analysis in a PRDM1 expressing cell type derived from the KO cells should be shown to confirm lack of PRDM1 protein in knockout cells (ideally WT and rescue cells would be used as controls). Sequencing data to show the disrupted locus would also normally be shown. How was the rescue performed? The only information I could find was that this was Prdm1 overexpression. This would not normally be tolerated by naïve cells (see PMID: 23851488 in which it is clearly stated that ES cells do not tolerate PRDM1 expression). I therefore find it difficult to understand what the GFP positive cells in Figure 5b and Supplementary Figure 5d are? Was this a conditional overexpression system to allow survival of these cells? Where stable Prdm1 overexpressing cell lines obtained?

Response: We apologize for the unclear statements. We generated the Prdm1-KO EpiSC, and detected the expression of PRDM1 protein (original supplementary Fig. 5b) at Day1 during BiPNT. Accordingly, we offered the western blot result (Rebuttal Fig. 6a or Supplementary Fig. 5e) and genotyping data (Rebuttal Fig. 6b, c or Supplementary Fig. 5a, b) (the sequencing data are too long to be shown) for Prdm1-KO cell line. The rescue experiments were performed by forced expression of Prdm1 with lentivirus in Prdm1-KO EpiSC to obtain stable expression of Prdm1 in KO EpiSC for BiPNT. Furthermore, PRDM1 can still be detected in some day8 rescued ΔPE-Oct4-GFP⁺ cells (Rebuttal Fig. 6d), indicating the generation of both PGCLC and naïve cells in the rescue system. We hope these new data and explanations can dispel the doubts of the reviewer.

Rebuttal Fig. 6

3. PGCLC experiments. The emergence of PGCLCs during BiPNT is overall quite well described and demonstrated. Most importantly the functional experiments show testes repopulation. As mentioned above Figure 3b should be presented with different headings that don't include arbitrary gene designations. Figure 3c and 3d seem very clear. However, subsequent experiments become somewhat muddled.

a) Figure 3f. Why are PGCLCs induced in GK15 media alone? This is a very surprising result. In contrast, the GK15+cytokines experiment does seem an important finding – this is the clearest demonstration that initiating BiPNT re-establishes germline competence of EpiSC. Perhaps a side-by-side experiment with untreated EpiSCs would be even more clear-cut?

Response: We thank the reviewer for these opinions. Actually, in this section, GK15 was only applied in the second stage of BiPNT. We speculated that the cells undergo stage 1 induction have already restored the germline competence, since they have up-regulated in PGC genes such as *Prdm1* and *Tfap2c*, whose over-expression can induce PGC fate in the absence of cytokines⁴. We also performed the experiment of PGCLC induction by GK15 or GK15+cytocines with untreated EpiSC, and show here that no BV⁺SV⁺ cells were detected (Rebuttal Fig.7 or Supplementary Fig. 3e).

Rebuttal Fig. 7

b) The CD105 experiments (for instance 3f) are difficult to interpret. The FACS plot seems to indicate that at least some EpiSCs will spontaneously upregulate this marker when plated in 2iL? Why not just perform an immunostaining for Klf4? This should very clearly show transition from a PGCLC state to an ESC state – and is a validated marker.

Response: We thank the reviewer for these questions. The CD105 experiment is the response to previous reviewer for interpreting the question about “capturing the transition from PGCLC to Naïve transition (P_GNT)”. In the revised manuscript, we offered the result of immunostaining for KLF4 as suggested by the reviewer (Rebuttal Fig. 8a, b or Fig. 3e, f), and removed the original CD105 data.

Rebuttal Fig. 8

c) H19 – although there is some loss of methylation, it is not at all clear that imprint erasure has occurred (and indeed this would not necessarily be expected in PGCLCs). Perhaps the authors are measuring the effect of 2iL to reduce DNA methylation more globally? As Day 6 BVSC cells have presumably been exposed to 2iL for 3 days?

Response: We agree with the reviewer that imprint erasure is not necessary for PGCLCs and 2iL can lead to global reduction of DNA methylation. Here, we also analyze the *H19* methylation state of BV⁺SC⁺ cells induced with GK15+cytokines instead of 2iL medium in the second stage of BiPNT. Similarly, we observed a reduced DNA methylation state at *H19* loci in GK15+cytokines induced BV⁺SC⁺ cells (Rebuttal Fig. 9). These data may indicate the occurrence of imprint erasure in BV⁺SC⁺ cells during BiPNT, consistent with previous report shown in PGCLC^{4,5}.

Rebuttal Fig. 9

d) **Chimera experiments.** More careful explanation of the findings is needed here. Although it is not consistently stated, it appears the injected BV+SC+ cells are from Day 6 cultures (meaning they have already been exposed to 2iL for 3 days? How exactly were in BV+SC+ cells passaged in 2i/L treated? On what day were they passaged and replated, and how much longer were they cultured for? How do the authors interpret the formation of a single chimera for the BV+SC+ cells? Is this due to contamination with naïve cells? Or due to rapid conversion of some PGCLCs to naïve pluripotency in their protocol (which does expose PGCLCs to 2iL).

Response: We thank for these issues and appreciate the careful reading of our manuscript by this reviewer and are glad to explain the detailed information about chimera experiments here. Just as mentioned by this reviewer, the BV⁺SC⁺ cells have already been exposed to 2i for 3 days (PNT day6). They were subsequently injected directly into receipt embryos for chimeras testing after sorting. On the other hand, some BV⁺SC⁺ cells were re-plated with 2iL medium for 4 days culture and further once-passaged culture (P1) for complete transition into naïve state. Afterward, these naïve like cells were harvested for chimeras testing. For the single chimera generated by BV⁺SC⁺ cells, we think is due to the rapid conversion of some PGCLCs to Naïve pluripotency, as we find that the Day 6 BV⁺SC⁺ cells in BiPNT show higher expression for pluripotency genes such as *Esrrb* comparing with other PGCs or PGCLCs (Rebuttal Fig. 10).

Rebuttal Fig. 10

Finally taking my points 2 and 3 together. If the BiPNT transition requires an obligate route through a PGCLC state – then does gating out all the BV+SC+ positive cells on Day 6 (and/or Day 8) completely prevent the formation of pluripotent stem cells clones? For instance, can the BVSC single or double negative cells in Figure 3f form naïve pluripotent stem cells (without themselves going through a BV+SC +positive state)?

Response: We thank the reviewer for this great advice. Accordingly, we performed the experiments as suggested, and show here that, the BV-SC⁻ cell sorted at day 6 could not form naïve pluripotent stem cells when further culturing in 2iL medium for 4 days as BV+SC⁺ cell did (Rebuttal Fig. 11a), in spite of a slight increasing in *Klf4* expression (Rebuttal Fig. 11b). These data further prove that the BiPNT requires an obligate route through a PGCLC state.

Rebuttal Fig. 11

3. Testes repopulation. This is important data and looks quite clear cut. Ideally an uninjected control should be included in Supplementary Figure 4. It seems quite clear that repopulation occurs, and I don't necessarily think further experiments are required. However, the authors should comment on whether they attempted to test the functionality

of any sperm/spermatids. Were any teratoma observed? (Perhaps indicating that early transition to naïve pluripotency does occur in the BiPNT protocol).

Response: We thanks the reviewer for these constructive suggestions. Accordingly, in the revised manuscript, we offered the uninjected control as suggested (Rebuttal Fig. 12 or Supplementary Fig. 4d). Indeed, we are trying to test the reproductive ability for the generated sperm/spermatids by mating the injected *W/W^v* mice to female mice, which may take quite a long time. And we haven't observed any teratoma formation.

Rebuttal Fig. 12

4. Mechanistic aspects. The model presented for the impact of DOT1L inhibition is clearly incomplete, but there are some interesting experiments and observations. This is already an extensive manuscript, and so I do not think that a complete mechanistic dissection is necessarily required. However, I think the authors should focus on more clearly explaining the experiments they have done, and emphasising what is and is not known, rather than trying to present the model as complete. For instance, it is not clear why DOT1L inhibition leads to upregulation of *Nanog* or *Tfap2c* and whether this is related to changes in H3K79methylation at these loci? A direct connection between alterations in expression of *Gata3/6* and of *Nanog* (and other PGC genes) in generating PGCLCs in BiPNT is really not clear. This is perhaps also due to the experiments being presented in a slightly idiosyncratic order. Also, it is not clear why the focus changes from *Gata3/6* to *Gata2*. If the main function of DOT1L inhibition is to reduce activation of *Gata* factors during BiPNT, then presumably knockout of these factors should reduce the requirement for DOT1L inhibition?

Response: We thank this reviewer for these suggestions. Indeed, although the expression of *Nanog* and *Tfap2c* are up-regulated by DOT1L inhibition, we observed a decrease of H3K79me2 at these loci (Rebuttal Fig. 13a or Fig. 6i). In the contrary, the down regulated genes such as *Gata3/6* are associated with the loss of H3K79me2 upon DOT1L inhibition (Rebuttal Fig. 13a or Fig. 6i), which is consistent with the role for H3K79me2 in active transcription¹. Rationally, we tend to the hypothesis that the function for DOT1L inhibition in PGC induction is due to the repression of those lineage factors such as *Gata3/6*. To test whether *Gata3/6* knockout can reduce the requirement for DOT1L inhibition, we generated *Gata3* or *Gata6* KO EpiSCs cell lines (Rebuttal Fig. 13b-e

or Supplementary Fig. 6g-j). And found that *Gata3/6* KO can promote the generation of BV⁺SC⁺ cells in the absence of DOT1Li (Rebuttal Fig. 13f, g or Fig. 6j, k). RT-qPCR further confirmed the up-regulation of PGC relevant genes such as *Nanog*, *Prdm1*, *Nanos3* and *Dppa3* in *Gata3/6* KO-day 6 cells (Rebuttal Fig. 13h or Supplementary Fig. 6k). In addition, the presence of DOT1Li can further enhance the efficiency of BV+SC+ cell induction in *Gata3* or *Gata6* KO EpiSCs, comparing with WT EpiSCs (Rebuttal Fig. 13g or Fig. 6k). These data indicate that *Gata3* and *Gata6* are DOT1Li targets during BV+SC+ cell induction from EpiSCs.

As for *Gata2*, it was predicted by using pySCENIC to be a regulon of trophoblast branch (Rebuttal Fig. 14a or Supplementary Fig. 7a). Indeed, we also find that *Gata2* is the downstream target of *Gata3*, as *Gata3* over-expression can upregulate the expression of *Gata2*, while *Gata3* KO leads to downregulation of *Gata2* (Rebuttal Fig. 14b). Furthermore, although *Gata2*-KO has no obvious effect in the generation of BV+SC+ cells in the absence of DOT1Li, it does promote the induction of BV+ cells (Rebuttal Fig. 14c). These are the reasons we focus on *Gata2* later.

Rebuttal Fig. 13

Rebuttal Fig. 14

5. EG cell-like cells (EGCLCs). Should the pluripotent stem cell lines derived using the BiPNT protocol actually be referred to as EGCLCs, given they apparently all go through a PGCLC state? Does Jak inhibition block the formation of naïve pluripotent stem cells/EGCLCs, as would be anticipated from ref 31 (PMID: 24052943)

Response: We thank the reviewer for these suggestions. Yes, the pluripotent stem cells here should be referred to EGCLCs, as the methylation state of the imprinted genes such as *Peg1*, *Peg3* and *Snrpn* in the derived naïve cells here are different from naïve ESCs (Rebuttal Fig.15a or Supplementary Fig. 5g), but similar to EGC⁶. In addition, the JAK inhibition can permit the induction of the Δ PE-*Oct4*-GFP⁺/KLF4⁻ cells (PGCLC), but not the Δ PE-*Oct4*-GFP⁺/KLF4⁺ naïve cells (Rebuttal Fig.15b, c or Supplementary Fig. 5h, i), indicating the block of transition from PGCLC to naïve pluripotent stem cells/EGCLCs by JAK inhibition.

Rebuttal Fig. 15

Minor points

1. Introduction. Line 62. This is not an adequate definition of pluripotency (the capacity to generate all the cells types in an individual must be the property of a single cell).

Response: We thank this reviewer for the mention of this point and have modified this sentence in the revised manuscript.

2. Introduction. Line 65-66. The division between ESC and EpiSC is confusing here. Are they really so similar by gene expression profile? Their in vitro differentiation potentials are also different – as typically EpiSCs do not efficiently give rise to PGCs. I would rework this section as it is a bit contradictory to the rest of the manuscript.

Response: We thank this reviewer for the mention of this mistake and have modified this sentence in the revised manuscript.

3. Line 70-71 – unclear English

Response: We have modified this sentence in the revised manuscript.

4. Line 81-82 – unclear English

Response: We have modified this sentence in the revised manuscript.

5. Line 133 – ‘Naïve ones’ – meaning unclear.

Response: We thank this reviewer for the mention of this point and have clarified the “naïve ones” to “naïve pluripotent stem cells” in the revised manuscript.

6. Line 145 – seems to include a change of font?

Response: We thank this reviewer for the very detailed revision of the manuscript, and have unified the font into Arial.

7. Line 162 – imprinted genes

Response: We thank this reviewer for pointing out this mistake and have modified in the revised manuscript.

8. Supplementary Figure 2c (and line 181-183). The big, flat cells shown are not obviously trophoblast giant cells (TGCs) and there is insufficient data to indicate successful generation of TGCs. I would just omit this – it really is not an important aspect of the paper. The exact trophoblast potential of these cells could surely form part of a future manuscript.

Response: We agree with this reviewer, this section was response to previous reviewer. We have removed this part in the revised manuscript.

9. Supplementary Figure 3e is difficult to interpret as it is performed on bulk cultures. In addition, the impact of these factors on PGC conversion to pluripotency is highly context dependent as shown by ref 31 (PMID: 24052943) and PMID: 32944903.

Response: We agree with the opinion of this reviewer. And we have removed this part in the revised manuscript.

10. Line 248-249 – incomplete sentence, please correct

Response: We have modified this sentence in the revised manuscript.

11. Line 301 – chimeras

Response: We have modified this mistake in the revised manuscript.

12. Line 311 – the authors have not shown the PGCLCs produce functional germ cells. This statement should be altered unless new data is available.

Response: We thank this reviewer for pointing out this mistake and have modified in the revised manuscript.

13. Line 458. Spelling of blastocyst

Response: We thank this reviewer for pointing out this mistake and have modified in the revised manuscript.

14. Zbtb7a/b. If this data is to be included, it should be in the Results section and properly incorporated into the narrative.

Response: We thank this reviewer for the suggestion, and we have removed this part for a shorter manuscript.

Reference:

1. Steger, D.J. *et al.* DOT1L/KMT4 recruitment and H3K79 methylation are ubiquitously coupled with gene transcription in mammalian cells. *Mol Cell Biol* **28**, 2825-2839 (2008).
2. Yoshimizu, T. *et al.* Germline-specific expression of the Oct-4/green fluorescent protein (GFP) transgene in mice. *Dev Growth Differ* **41**, 675-684 (1999).
3. Bao, S. *et al.* Epigenetic reversion of post-implantation epiblast to pluripotent embryonic stem cells. *Nature* **461**, 1292-1295 (2009).
4. Nakaki, F. *et al.* Induction of mouse germ-cell fate by transcription factors in vitro. *Nature* **501**, 222-226 (2013).
5. Hayashi, K., Ohta, H., Kurimoto, K., Aramaki, S. & Saitou, M. Reconstitution of the mouse germ cell specification pathway in culture by pluripotent stem cells. *Cell* **146**, 519-532 (2011).
6. Gillich, A. *et al.* Epiblast stem cell-based system reveals reprogramming synergy of germline factors. *Cell Stem Cell* **10**, 425-439 (2012).

REVIEWERS' COMMENTS

Reviewer #4 (Remarks to the Author):

The authors have made a number of alterations that have improved the manuscript. However, there remain some issues that remain to be clarified/rectified.

1. DOT1Li

The explanation of the findings here remains unclear. It does make sense that loss of H3K79 methylation from the Gata genes leads to their downregulation. However, the authors start their explanation by stating 'RT-qPCR analysis further validates the activation of PGC markers such as Prdm1, Dppa3, Nanos3 and Prdm14 by DOT1L inhibitors (DOT1Li)' (line 312-314). This finding is never adequately explained, and the authors just make statements like 'restoring PGCC competence' (line 318) and 'activating PGC regulators' (377)

They perform RNA-seq, ATAC-seq and ChIP-seq after 3 days plus/minus DOT1L inhibition – however, this is not informative as to the mechanism of gene activation – as there is ample time for secondary effects. Are the authors suggesting that H3K79 methylation can be a repressive mark in certain contexts? They mention that for repressed genes H3K79 methylation is replaced by H3K27me3, but not at activated genes. But then the obvious question is what determines this? I mentioned in my previous review that I was not certain that a complete mechanistic dissection is necessary – however, equally it does not really work to present seemingly contradictory data without an explanation or hypothesis as to how this is happening. It remains completely unclear why loss of treatment with DOT1L inhibitor can activate some genes but repress others. However, as secondary effects have not been ruled out, the statement 'Interestingly, the loss of H3K79me2 by DOT1L inhibition leads to both gene repression (such as Gata3/6) and activation (such as Nanog, Tfap2c)' is not appropriate.

2. BMP treatment.

It is surprising that GK15 alone is sufficient to allow emergence of PGCLCs. Do the authors think this is because the whole BiPNT process is triggered by BMP4? More generally the fact that BMP both triggers BiPNT and is the key growth factor driving PGC specification (and PGCLC induction) should be discussed, as presumably this is key to the underlying mechanism. Addition of BMP is also different to other protocols for primed to naïve conversion, including those (such as rESCs) that have been proven not to require Prdm1 or transition through a PGC-like intermediate.

3. Minor issues

- Summary: 'clarity'. Sometimes bulk approaches do bring a great deal of clarity, and conversely single cell analysis often adds little. I think 'resolution' might be a better choice here.
- Summary 'reveals a new cell fate dynamics'. I am not sure this statement is grammatically correct. In addition, it is rather an alternative route (from primed to naïve state) that is revealed here, rather than anything in particular about the dynamics.
- Line 60-61: 'despite both refer to pluripotency'. Please revise English
- Line 120. 'hence'. 'thereafter' might be a better choice.
- Line 137 - 'ExE. ectoderm'. ExE should be defined and I am not convinced the period ('.') is needed here, and in the other examples.
- Line 140: 'cells apparent not express'. Please revise English
- Line 177: 'Obviously'. I would advise deleting this.
- Line 291 'Embryonic germ cell or EGC, another naive pluripotent cell lines originate from PGC, differs with ESC in imprinted pattern'. Some EG cell lines do exhibit loss of imprinting. However, so do many ES cells lines (Humpherys et al. Science 293, 95–97 (2001)). In addition, some EG cell lines exhibit normal imprints (Leitch et al. Nat Struct Mol Biol 20, 311–316 (2013)). Therefore this is not a reliable marker of germline derived cells.
- Figure 2J – 'unsorting'. Should perhaps read 'unsorted'

REVIEWERS' COMMENTS

Reviewer #4 (Remarks to the Author):

The authors have made a number of alterations that have improved the manuscript. However, there remain some issues that remain to be clarified/rectified.

1. DOT1Li

The explanation of the findings here remains unclear. It does make sense that loss of H3K79 methylation from the Gata genes leads to their downregulation. However, the authors start their explanation by stating 'RT-qPCR analysis further validates the activation of PGC markers such as Prdm1, Dppa3, Nanos3 and Prdm14 by DOT1L inhibitors (DOT1Li)' (line 312-314). This finding is never adequately explained, and the authors just make statements like 'restoring PGCC competence' (line 318) and 'activating PGC regulators' (377)

They perform RNA-seq, ATAC-seq and CHIP-seq after 3 days plus/minus DOT1L inhibition – however, this is not informative as to the mechanism of gene activation – as there is ample time for secondary effects. Are the authors suggesting that H3K79 methylation can be a repressive mark in certain contexts? They mention that for repressed genes H3K79 methylation is replaced by H3K27me3, but not at activated genes. But then the obvious question is what determines this? I mentioned in my previous review that I was not certain that a complete mechanistic dissection is necessary – however, equally it does not really work to present seemingly contradictory data without an explanation or hypothesis as to how this is happening. It remains completely unclear why loss of treatment with DOT1L inhibitor can activate some genes but repress others. However, as secondary effects have not been ruled out, the statement 'Interestingly, the loss of H3K79me2 by DOT1L inhibition leads to both gene repression (such as Gata3/6) and activation (such as Nanog, Tfap2c)' is not appropriate.

Response : We appreciate the reviewer for these thoughtful and valuable comments.

H3K79 methylation is generally considered as an active signal for transcription. However, in some cases, DOT1L and H3K79 methylation have been linked to transcriptional repression, like in mouse adrenal cells[1], cerebral cortex[2] or *Caenorhabditis elegans* [3], suggesting a context dependent manner for H3K79 methylation on gene expression. Therefore, in BiPNT system, the down- or up-regulation of certain gene associated with loss of H3K79 methylation may indicate the active and repressive role for H3K79 methylation on transcription, which may depend on other epigenetic modification or chromatin status at specific gene loci[2, 4, 5]. However, we agree with the reviewer that the premise of such hypothesis is to exclude the secondary effect of DOT1Li on the gene activation. To this end, we analyzed the transcription level for Day3-DOT1Li up-regulated genes (PGC genes: *Prdm1*, *Nanog*, *Tfap2c*) and down-regulated genes (*Gata3/6*) at the early stage (Day1) of BiPNT, respectively, and showed that, the expression of *Gata3/6* is repressed significantly by DOT1Li at Day1, while the expression of PGC genes hasn't obviously changed (Figure

1). In addition, *Gata3/6* deficiency can upregulate PGC genes and promote the induction of PGCLC in the absence of DOT1Li (Fig. 6j, k and Supplementary Fig. 6k of the manuscript). Taken together, these data indicate a secondary effect may exist for the activation of PGC genes by DOT1Li. Thus, we would like to modify our claim to that DOT1Li promote the activation of PGC genes and generation of PGCLC in part by facilitating the loss of H3K79me2 from *Gata3/6* and repressing their expression. We have modified our claim in the revised manuscript (including the result and discussion sections) , and hope the reviewer and editor agree with us on this point.

Figure 1

2. BMP treatment.

It is surprising that GK15 alone is sufficient to allow emergence of PGCLCs. Do the authors think this is because the whole BiPNT process is triggered by BMP4? More generally the fact that BMP both triggers BiPNT and is the key growth factor driving PGC specification (and PGCLC induction) should be discussed, as presumably this is key to the underlying mechanism. Addition of BMP is also different to other protocols for primed to naïve conversion, including those (such as rESCs) that have been proven not to require *Prdm1* or transition through a PGC-like intermediate.

Response : We thank the reviewer for these comments. Indeed, unlike other primed to naïve conversion systems, BMP4 is indispensable for the progress of BiPNT, as our previous data shown[6]. In addition, the emergence of PGCLCs in BiPNT is also BMP4 dependent (data not shown). Therefore, the induction of PGCLC by GK15 alone, instead of 2iL at stage 2 of BiPNT, should also attributed to the BMP4 treatment at stage 1. BMP4 is also the key factor for driving PGC specification, and PGC shares many markers with naïve pluripotent cells (Such as *Nanog*, *Prdm14*, *Klf2*, *Dppa3* etc.). Furthermore, we previously showed that BMP4 is able to activate naïve or PGC related genes such as *Klf2*, *Esrrb*, *Tfap2c* and *Dppa3* in BiPNT[6]. These data demonstrate that BMP4 could trigger both BiPNT and PGC specification, thus provided a link between PGCLC and BiPNT process. According to the reviewer's suggestion, we have discussed more clearly and made corresponding statements in the revised manuscript.

3. Minor issues

- Summary: 'clarity'. Sometimes bulk approaches do bring a great deal of clarity, and conversely single cell analysis often adds little. I think 'resolution' might be a better choice here.

Response: Thank the reviewer for this suggestion, we have revised it in the revised manuscript.

- Summary 'reveals a new cell fate dynamics'. I am not sure this statement is grammatically correct. In addition, it is rather an alternative route (from primed to naïve state) that is revealed here, rather than anything in particular about the dynamics.

Response: Thank the reviewer for this valuable suggestion, we have revised it in the revised manuscript.

- Line 60-61: 'despite both refer to pluripotency'. Please revise English

Response: Thank the reviewer and we have revised it as suggested.

- Line 120. 'hence'. 'thereafter' might be a better choice.

Response: Thank the reviewer and we have revised it as suggested.

- Line 137 - 'ExE. ectoderm'. ExE should be defined and I am not convinced the period ('.') is needed here, and in the other examples.

Response: Thank the reviewer and we have revised it as suggested.

- Line 140: 'cells apparent not express'. Please revise English

Response: Thank the reviewer for this comment, we have revised it in the revised manuscript.

- Line 177: 'Obviously'. I would advise deleting this.

Response: Thank the reviewer and we have revised it as suggested.

- Line 291 'Embryonic germ cell or EGC, another naive pluripotent cell lines originate from PGC, differs with ESC in imprinted pattern'. Some EG cell lines do exhibit loss of imprinting. However, so do many ES cells lines (Humpherys et al. Science 293, 95–97 (2001)). In addition, some EG cell lines exhibit normal imprints (Leitch et al. Nat Struct Mol Biol 20, 311–316 (2013)). Therefore this is not a reliable marker of germline derived cells.

Response: We thank the reviewer for reminding this point. Accordingly, in the revised manuscript, we removed this sentence, and only presented our data.

- Figure 2J – 'unsorting'. Should perhaps read 'unsorted'

Response: Thank the reviewer and we have revised it as suggested.

References

1. Zhang, W., et al., *Dot1a-AF9 complex mediates histone H3 Lys-79 hypermethylation and*

- repression of ENaC α in an aldosterone-sensitive manner.* J Biol Chem, 2006. **281**(26): p. 18059-68.
2. Franz, H., et al., *DOT1L promotes progenitor proliferation and primes neuronal layer identity in the developing cerebral cortex.* Nucleic Acids Res, 2019. **47**(1): p. 168-183.
 3. Cecere, G., et al., *The ZFP-1(AF10)/DOT-1 complex opposes H2B ubiquitination to reduce Pol II transcription.* Mol Cell, 2013. **50**(6): p. 894-907.
 4. Ferrari, F., et al., *DOT1L-mediated murine neuronal differentiation associates with H3K79me2 accumulation and preserves SOX2-enhancer accessibility.* Nat Commun, 2020. **11**(1): p. 5200.
 5. Wille, C.K. and R. Sridharan, *DOT1L inhibition enhances pluripotency beyond acquisition of epithelial identity and without immediate suppression of the somatic transcriptome.* Stem Cell Reports, 2022. **17**(2): p. 384-396.
 6. Yu, S., et al., *BMP4 resets mouse epiblast stem cells to naive pluripotency through ZBTB7A/B-mediated chromatin remodelling.* Nat Cell Biol, 2020. **22**(6): p. 651-662.